# Celsr1 adhesive interactions mediate the asymmetric organization of planar polarity complexes

**Sara N Stahley, Lena P Basta, Rishabh Sharan, Danelle Devenport***

Department of Molecular Biology, Princeton University, Princeton, United States

**Abstract** To orchestrate collective polarization across tissues, planar cell polarity (PCP) proteins localize asymmetrically to cell junctions, a conserved feature of PCP that requires the atypical cadherin Celsr1. We report that mouse Celsr1 engages in both *trans*- and *cis*-interactions, and organizes into dense and highly stable punctate assemblies. We provide evidence suggesting that PCP-mutant variant of Celsr1, Celsr1$^{Crsh}$, selectively impairs lateral *cis*-interactions. Although Celsr1$^{Crsh}$ mediates cell adhesion in trans, it displays increased mobility, diminishes junctional enrichment, and fails to engage in homophilic adhesion with the wild-type protein, phenotypes that can be rescued by ectopic *cis*-dimerization. Using biochemical and super-resolution microscopy approaches, we show that although Celsr1$^{Crsh}$ physically interacts with PCP proteins Frizzled6 and Vangl2, it fails to organize these proteins into asymmetric junctional complexes. Our results suggest mammalian Celsr1 functions not only as a *trans*-adhesive homodimeric bridge, but also as an organizer of intercellular Frizzled6 and Vangl2 asymmetry through lateral, *cis*-interactions.

## Introduction

Planar cell polarity (PCP) refers to the collective polarization of cells along a tissue plane and is controlled by a conserved set of membrane-associated proteins known as the core PCP pathway (*Yang and Mlodzik, 2015*; *Strutt et al., 2016a*; *Butler and Wallingford, 2017*). The PCP pathway directs a diverse array of polarized cell behaviors across a variety of tissues including directed ciliary beating, collective cell motility, and body hair alignment across the skin surface (*Seifert and Mlodzik, 2007*; *Niessen et al., 2012*; *Devenport, 2016*; *Goffinet and Tissir, 2017*; *Ebnet et al., 2018*). The PCP pathway is essential for embryonic development, and genetic disruption of PCP leads to severe developmental defects, including neural tube and congenital heart defects (*Kibar et al., 2001*; *Curtin et al., 2003*; *Phillips et al., 2007*; *Qiao et al., 2016*; *Butler and Wallingford, 2017*; *Wang et al., 2019*). To align polarity across a tissue, PCP proteins interact intercellularly to couple polarity between neighboring cells, but the mechanisms regulating these molecular interactions remain unknown (*Ebnet et al., 2018*).

A defining feature of PCP is the asymmetric localization of membrane-associated PCP proteins at cell junctions (*Yang and Mlodzik, 2015*; *Butler and Wallingford, 2017*). The direction of polarity is thought to be determined by a global cue that biases PCP localization relative to the tissue axes (*Yang and Mlodzik, 2015*; *Aw and Devenport, 2017*; *Butler and Wallingford, 2017*). PCP proteins then redistribute from initially uniform localizations to become enriched at cell junctions oriented along one tissue axis (*Devenport, 2014*; *Aw and Devenport, 2017*; *Ebnet et al., 2018*). Transmembrane proteins Frizzled (Fz) and Van Gogh-like (Vangl) localize to opposing sides of the junction, where they interact intercellularly in complex with atypical cadherin Celsr1. Cytoplasmic proteins Dishevelled and Prickle colocalize with Fz and Vangl, respectively, and are important for amplifying asymmetry through oligomerization domains that mediate self-recruitment, and by promoting mobility and disassembly of the oppositely oriented complex (*Devenport, 2014*; *Butler and Wallingford,*

*For correspondence:
danelle@princeton.edu

2017; *Harrison et al., 2020*). How Celsr1-mediated intercellular interactions contribute to this asymmetric reorganization of PCP protein complexes is poorly understood.

Celsr1 is a large atypical cadherin with an extracellular domain comprised of nine N-terminal cadherin repeats followed by several EGF and laminin repeats (*Wang et al., 2014*; *Goffinet and Tissir, 2017*). The *Drosophila* homolog Flamingo (Fmi; also Starry night, Stan) engages in homophilic adhesion (*Usui et al., 1999*) and acts upstream of the other core PCP proteins (*Chae et al., 1999*; *Bastock et al., 2003*; *Lawrence et al., 2004*) where it is required for their stable localization to junctions (*Axelrod, 2001*; *Feiguin et al., 2001*; *Shimada et al., 2001*; *Tree et al., 2002*; *Bastock et al., 2003*; *Strutt and Strutt, 2007*; *Chen et al., 2008*; *Strutt and Strutt, 2008*; *Strutt et al., 2016a*). Although the cadherin family of proteins are known to engage in both *trans-* and *cis*-interactions to mediate adhesion (*Niessen et al., 2011*; *Brasch et al., 2012*; *Priest et al., 2017*; *Honig and Shapiro, 2020*), little is known about the Celsr1 adhesive interface, or how its adhesive interactions promote PCP asymmetry (*Brasch et al., 2012*; *Wang et al., 2014*; *Krishnan et al., 2016*). The *Crash* (*Celsr1$^{Crsh}$*) mutant mouse, which harbors a single amino acid substitution (D1040G) between extracellular cadherin (EC) repeats 7 and 8 of Celsr1 (*Curtin et al., 2003*), provides an inroad into understanding cadherin-mediated adhesion in the establishment of PCP. Originally identified in a chemical mutagenesis screen, mice homozygous for the *Celsr1$^{Crsh}$* allele display severe defects associated with a loss of PCP: a complete failure to close the neural tube, misoriented stereocilia bundles in the inner ear, and misalignment of hair follicles across the skin surface (*Curtin et al., 2003*; *Devenport and Fuchs, 2008*). A point mutation in the cadherin repeats of Celsr1 is predicted to affect homophilic adhesion, but not necessarily protein levels or intracellular transport, allowing us to selectively address the function of Celsr1 adhesive interactions in PCP.

Here, we provide evidence that the membrane proximal cadherin repeats of Celsr1 modulate its adhesive function to regulate PCP dynamics. Specifically, we use a combination of cell adhesion and biochemical assays, as well as live and super-resolution imaging to show that the *Celsr1$^{Crsh}$* mutation in Celsr1 does not eliminate *trans*-adhesive interactions but rather interferes with lateral *cis*-interactions that organize PCP proteins into stable, clustered assemblies. Importantly, Celsr1$^{Crsh}$ is capable of physically interacting with both Fz6 and Vangl2, yet fails to organize these proteins into asymmetric junctional complexes in vivo. We propose that *cis*-interactions are a novel facet of Celsr1 adhesion, essential for the molecular organization of asymmetric PCP complexes.

## Results

### The *Celsr1$^{Crsh}$* mutation abolishes Celsr1 asymmetric localization and acts as a partial dominant-negative

To investigate how extracellular adhesive interactions contribute to PCP asymmetry, we examined polarity establishment in *Celsr1$^{Crsh/Crsh}$* mutant embryos (*Figure 1A–D*). In basal epithelial cells of the epidermis, PCP proteins accumulate preferentially at anterior–posterior junctions and are depleted from mediolateral junctions (*Devenport and Fuchs, 2008*; *Aw et al., 2016*). For simplicity, we refer to these as 'vertical' and 'horizontal' junctions, respectively (*Figure 1A*). Celsr1 asymmetry arises progressively between embryonic day E12.5 and E15.5 of development where both the magnitude and collective alignment of polarity increases (*Devenport and Fuchs, 2008*; *Aw et al., 2016*). To quantify Celsr1 asymmetry in *Celsr1$^{Crsh/Crsh}$* mutant epidermis, we used automated segmentation and calculated the nematic order of Celsr1 fluorescence intensity. In agreement with our previous qualitative analysis, Celsr1 enrichment at vertical junctions was completely lost in E15.5 *Celsr1$^{Crsh/Crsh}$* embryos with Celsr1 localizing uniformly to both vertical and horizontal cell borders (*Figure 1C and D*; *Devenport and Fuchs, 2008*). Heterozygous *Celsr1$^{Crsh/+}$* embryos also displayed a polarity defect, where both the magnitude and collective alignment of Celsr1 asymmetry were reduced compared to *Celsr1$^{+/+}$* littermates (*Figure 1C and D*). Thus, a D1040G substitution in the cadherin repeats of Celsr1 abolishes its ability to localize asymmetrically and partially interferes with Celsr1 function in a dominant manner.

Interestingly, the loss of Celsr1 asymmetry observed in *Celsr1$^{Crsh/Crsh}$* mutants is comparable to that caused by mutations that eliminate Fz6 or Vangl2 protein or prevent Vangl2 trafficking (*Guo et al., 2004*; *Torban et al., 2007*; *Devenport and Fuchs, 2008*; *Merte et al., 2010*; *Yin et al., 2012*; *Cetera et al., 2017*). By contrast, *Celsr1$^{Crsh}$* strongly impairs PCP establishment without

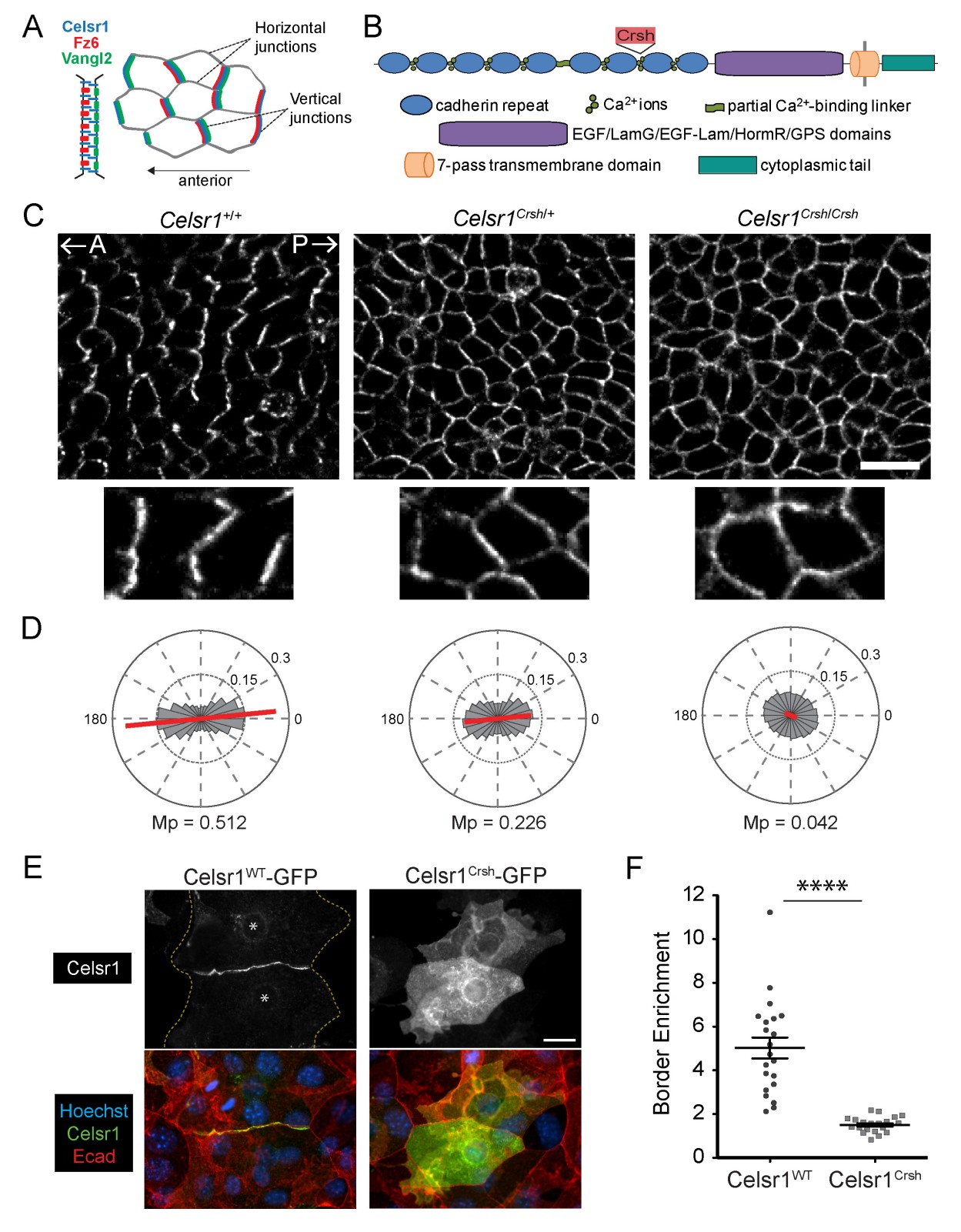

**Figure 1.** Planar cell polarity (PCP)-disrupting mutation *Celsr1Crsh* perturbs Celsr1 polarity in the epidermis and reduces border enrichment in keratinocytes. (A) Schematics of core transmembrane PCP proteins at a cell–cell junction and their polarized localization at vertical junctions in the epidermis. (B) Schematic of Celsr1 protein domain organization with emphasis on the extracellular cadherin repeats. Non-cadherin repeat domains shown as collective purple region. Not drawn to scale. Location of *Crash* (*Celsr1Crsh*, D1040G) mutation indicated by red text box. (C) Planar confocal

*Figure 1 continued on next page*

*Figure 1 continued*

section through the basal layer of whole-mount back skin from E15.5 *Celsr1^{+/+}*, *Celsr1^{Crsh/+}*, and *Celsr1^{Crsh/Crsh}* embryos stained for Celsr1. A, anterior; P, posterior. Scale bar, 20 μm. (D) Quantification of Celsr1 polarity. Mp indicates the magnitude of polarity. *Celsr1^{+/+}*, five embryos (9010 cells); *Celsr1^{Crsh/+}*, six embryos (11,614 cells); *Celsr1^{Crsh/Crsh}*, five embryos (13,892 cells). (E) Cultured mouse keratinocytes transfected with Celsr1^{WT}-GFP or Celsr1^{Crsh}-GFP and switched to high calcium for 24 hr. For the cell pair expressing Celsr1^{WT}-GFP, yellow dashed lines indicate outline of cells and * denotes location of nuclei. Scale bar, 20 μm. (F) Celsr1 border enrichment quantification (two-tailed unpaired *t*-test with Welch's correction, ****p<0.0001; n = 21 cell pairs).

The online version of this article includes the following source data and figure supplement(s) for figure 1:

**Source data 1.** Data accompanying *Figure 1*.

**Figure supplement 1.** Celsr1^{Crsh} does not enrich at keratinocyte cell borders when co-expressed with Celsr1^{WT} and acts as a partial dominant negative.

significantly altering Celsr1 trafficking or cell surface levels, or affecting expression of other core PCP proteins (*Devenport and Fuchs, 2008*; *Murdoch et al., 2014*). Thus, *Celsr1^{Crsh}* is a unique PCP mutation that allows us to specifically address the function of Celsr1 adhesion in PCP.

## Celsr1^{WT} and mutant Celsr1^{Crsh} mediate homotypic adhesion in trans

The position of the *Celsr1^{Crsh}* mutation between EC7 and EC8 suggested that it may inhibit homotypic adhesion of Celsr1. Consistent with previous work, transfected Celsr1^{Crsh}-GFP failed to enrich at cell–cell interfaces between neighboring keratinocytes in contrast to the strong border enrichment of wild-type (WT) Celsr1 (Celsr1^{WT}-GFP) (*Figure 1E and F*; *Devenport and Fuchs, 2008*). When Celsr1^{WT}-GFP and Celsr1^{Crsh}-GFP were co-expressed in keratinocytes, enrichment of Celsr1^{WT} at junctions was reduced (*Figure 1—figure supplement 1A–C*), supporting the idea that the *Celsr1^{Crsh}* mutation interferes with Celsr1 adhesion in a dominant-negative manner (*Qu et al., 2010*; *Murdoch et al., 2014*).

To directly test whether *Celsr1^{Crsh}* impairs Celsr1 adhesion, we established a cell aggregation assay to determine if Celsr1 alone is sufficient to mediate cell–cell adhesion. K-562 cells, which are cadherin-free, non-adhesive, and grow in suspension (*Ozawa and Kemler, 1998*; *Schreiner and Weiner, 2010*; *Baykal-Köse et al., 2020*), were stably transfected with Celsr1 variants and assayed for their ability to induce cell aggregation, an indicator of cell adhesion activity. Celsr1^{WT}-GFP expressing K-562 cells formed aggregates similar to positive control cells expressing a chimeric protein containing the extracellular domain of E-cadherin. In contrast, negative control cells expressing GFP alone remained as a single cell suspension (*Figure 2A–C*). When mixed with cells expressing the E-cadherin chimera, Celsr1^{WT} expressing cells did not co-aggregate, but rather sorted into distinct clusters, demonstrating Celsr1 mediates homophilic binding (*Figure 2D*). Homophilic adhesion was dependent on the Celsr1 extracellular domain as cells expressing a Celsr1 variant lacking the extracellular domain (Celsr1^{ΔN}) failed to aggregate and also failed to co-aggregate with cells expressing full-length Celsr1 (*Figure 2E and F*). Like classical cadherins, Celsr1-mediated aggregation was calcium-dependent (*Figure 2G and H*). These results confirm that Celsr1 mediates homophilic, calcium-dependent cell adhesion in trans.

Surprisingly, K-562 cells expressing Celsr1^{Crsh}-GFP formed aggregates comparable to Celsr1^{WT}-GFP expressing cells, demonstrating the *Celsr1^{Crsh}* mutation does not impair the ability of Celsr1^{Crsh} to mediate adhesion in trans (*Figure 2I and J*). To test whether the Celsr1^{Crsh} variant could mediate homophilic adhesion with the WT protein, we performed mixing assays between cells expressing either Celsr1^{WT}-GFP or Celsr1^{Crsh}-mCherry. Unlike controls where Celsr1^{WT}-GFP cells co-aggregated with Celsr1^{WT}-mCherry cells, Celsr1^{Crsh}-mCherry expressing cells failed to co-aggregate with Celsr1^{WT}-GFP and sorted out into distinct cell clusters (*Figure 2K and L*, *Figure 2—figure supplement 1A*). This aggregate sorting behavior is likely specific to the effect of the *Celsr1^{Crsh}* mutation and not due to differences in expression or cell surface localization as cell surface expression of Celsr1^{Crsh} was similar to Celsr1^{WT} (*Figure 2—figure supplement 1C and D*). Interestingly, at early timepoints post mixing, Celsr1^{Crsh}-mCherry and Celsr1^{WT}-GFP cells did adhere, but later separated into distinct, unmixed aggregates (*Figure 2M and N*, *Figure 2—figure supplement 1B*). In support of these results, Celsr1^{Crsh}-mCherry and Celsr1^{WT}-GFP also failed to interact in trans at cell borders in mixed keratinocytes (*Figure 2—figure supplement 2*). These findings indicate that Celsr1^{Crsh} is a novel Celsr1 variant with altered homophilic binding preferences that fails to form stable *trans*-adhesive interactions with the WT protein.

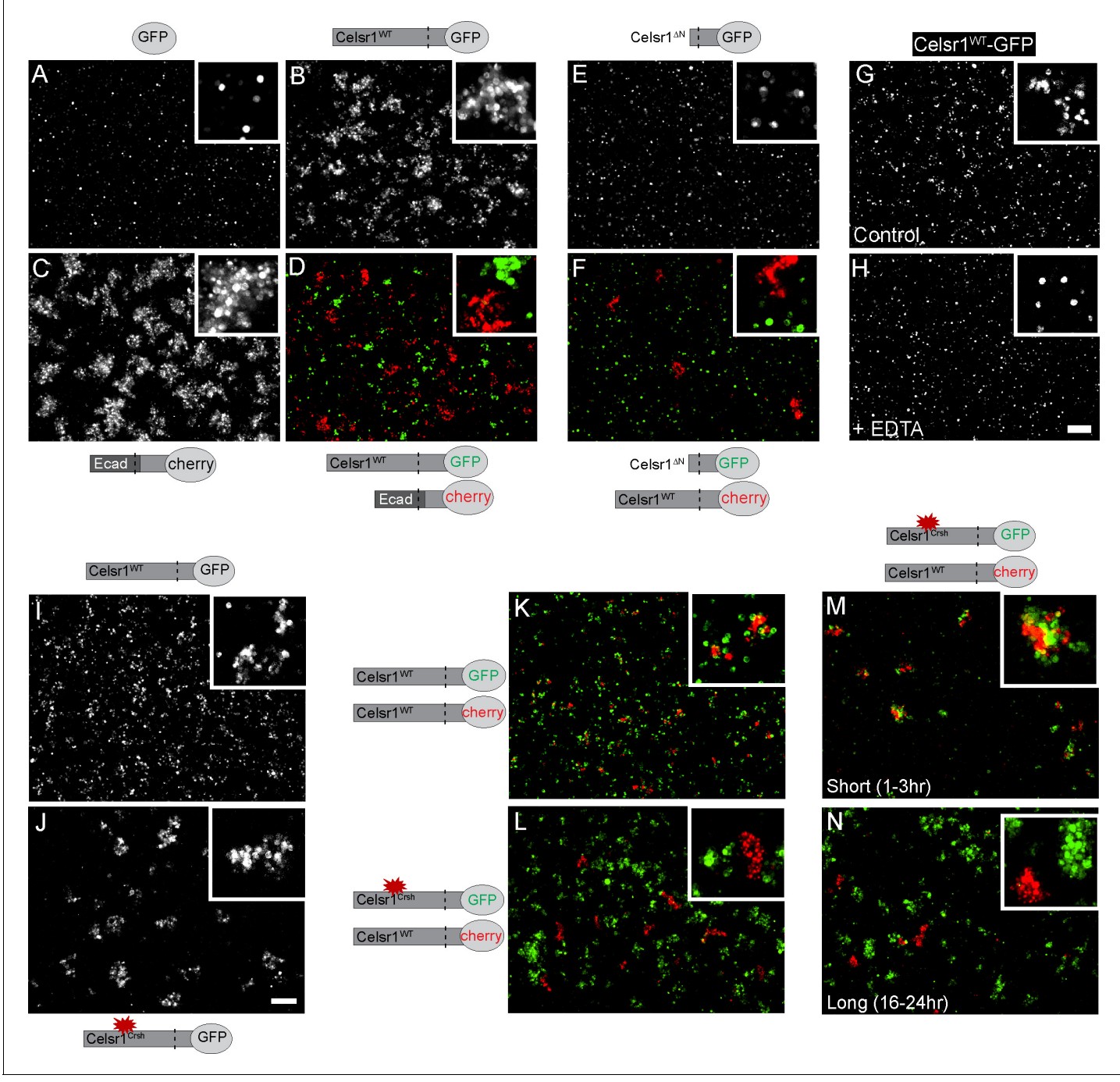

**Figure 2.** Celsr1WT and Celsr1Crsh mediate cell adhesion but do not form stable homotypic contacts between cells. Cell aggregation assay. K-562 cells stably expressing protein variants indicated were assayed for their ability to induce cell aggregation, or cell adhesion. (A–D) Cells expressing GFP (A), Celsr1WT-GFP (B), EcadECD+TM-Celsr1CT-mCherry (C) or Celsr1WT-GFP mixed with EcadECD+TM-Celsr1CT-mCherry (D) imaged live following a 1.5 hr aggregation. Ecad, E-cadherin; ECD, extracellular domain; TM, transmembrane domain; CT, cytoplasmic tail. (E and F) Cells expressing Celsr1ΔN-GFP (E) or Celsr1ΔN-GFP mixed with Celsr1-mCherry (F) imaged fixed following a 3 hr aggregation. ΔN, lacking nearly the entire N-terminal extracellular domain. (G and H) Celsr1WT-GFP expressing cells incubated in either media alone or media containing 10 mM EDTA (calcium chelator) during a 2 hr aggregation. (I and J) K-562 cells expressing Celsr1WT or Celsr1Crsh mediate adhesion by the formation of cell aggregates. (K and L) Celsr1WT-GFP and Celsr1Crsh-GFP expressing K-562 cells were mixed for 16–24 hr with Celsr1WT-mCherry expressing cells. (M and N) Mixed cell aggregation assay between K-562 cells expressing either Celsr1Crsh-GFP or Celsr1WT-mCherry for incubation periods indicated (short, 1–3 hr or long, 16–24 hr). Scale bar, 200 µm.

The online version of this article includes the following source data and figure supplement(s) for figure 2:

*Figure 2 continued on next page*

*Figure 2 continued*

**Source data 1.** Data accompanying *Figure 2*.
**Figure supplement 1.** Celsr1Crsh does not form stable *trans*-interactions with Celsr1WT in K-562 cells.
**Figure supplement 2.** Celsr1Crsh does not form stable *trans*-interactions with Celsr1WT in keratinocytes.

## The ability of Celsr1 to concentrate and cluster at intercellular junctions is diminished by *Celsr1Crsh*

Cadherin family members typically engage in two types of interactions to mediate cell–cell adhesion, intercellular *trans*-interactions between cadherins on opposing cells and lateral *cis*-interactions within the same cell (*Brasch et al., 2012*; *Priest et al., 2017*; *Honig and Shapiro, 2020*). *Trans*- and *cis*-interactions are thought to cooperate, functioning together to contribute to junction assembly, stability, and overall adhesive strength (*Zhang et al., 2009*; *Wu et al., 2010*; *Harrison et al., 2011*; *Niessen et al., 2011*; *Brasch et al., 2012*; *Priest et al., 2017*). Given that the Celsr1Crsh mutant was still able to mediate adhesion in trans, we hypothesized that the *Celsr1Crsh* mutation may impair Celsr1 interactions in cis. Consistently, high resolution confocal imaging of K-562 cell aggregates revealed stark differences in the subcellular localization of Celsr1WT versus Celsr1Crsh. Whereas Celsr1WT-GFP strongly enriched at interfaces connecting neighboring cells (*Figure 3A*, open arrows) and was depleted from non-contacting edges (closed arrows), Celsr1Crsh-GFP was distributed diffusely and broadly across the cell surface, localizing to both contacting (open arrows) and non-contacting cell edges (closed arrows). Quantifying the ratio of Celsr1-GFP fluorescence intensity at contacting versus non-contacting surfaces revealed a significant reduction in border enrichment (*Figure 3B*), suggesting that *Celsr1Crsh* disrupts the ability of Celsr1 to concentrate at intercellular junctions.

In *Drosophila*, core PCP proteins organize into punctate assemblies along cell junctions (*Strutt and Strutt, 2009*; *Strutt et al., 2011*). By both confocal and super-resolution structured illumination microscopy (SIM), we observe that Celsr1 is similarly organized into puncta at cell borders in mouse epidermis and keratinocytes (*Devenport and Fuchs, 2008*; *Aw et al., 2016*; *Figure 4A*,

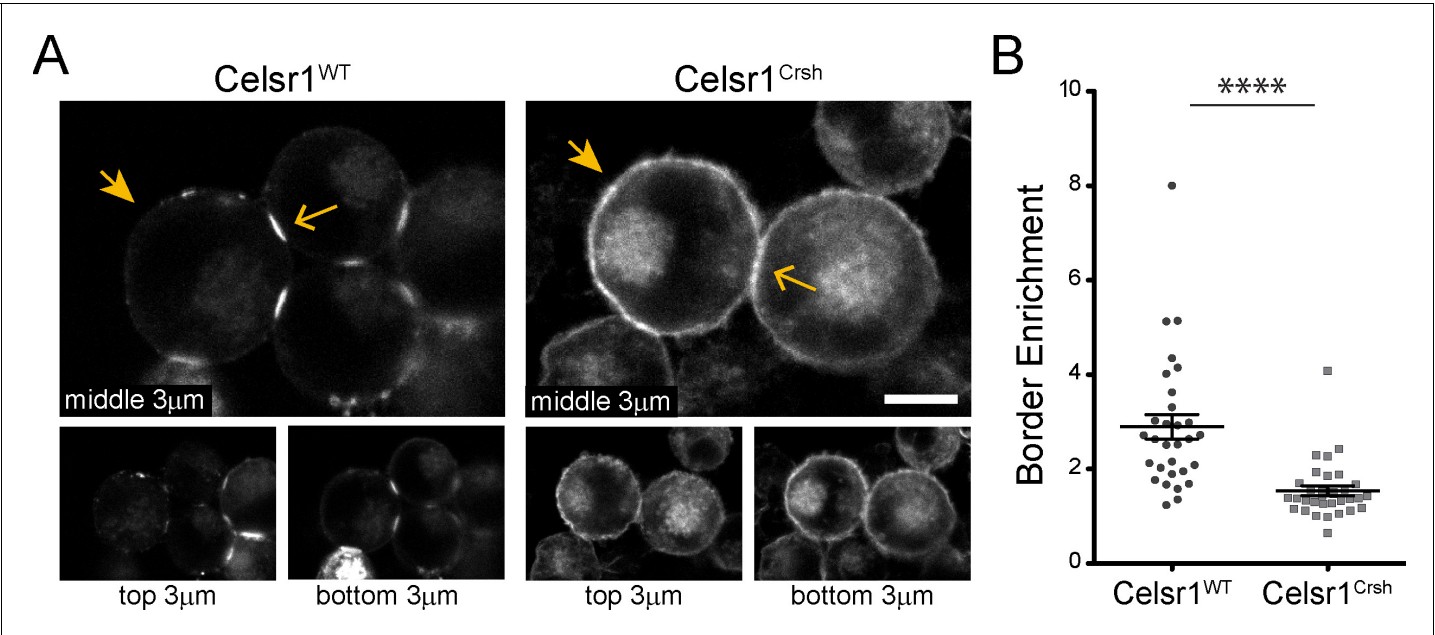

**Figure 3.** Celsr1Crsh displays altered surface distribution and border enrichment in K-562 cells. (**A**) Confocal max projections (3 µm each from top, middle and bottom of cell aggregate) of GFP tagged Celsr1WT and Celsr1Crsh aggregates. Scale bar, 10 µm. (**B**) Border enrichment quantification (Celsr1Crsh border enrichment reduced 46.8%; two-tailed unpaired *t*-test with Welch's correction, ****p<0.0001; Celsr1WT, n = 30; Celsr1Crsh, n = 33). The online version of this article includes the following source data for figure 3:

**Source data 1.** Data accompanying *Figure 3*.

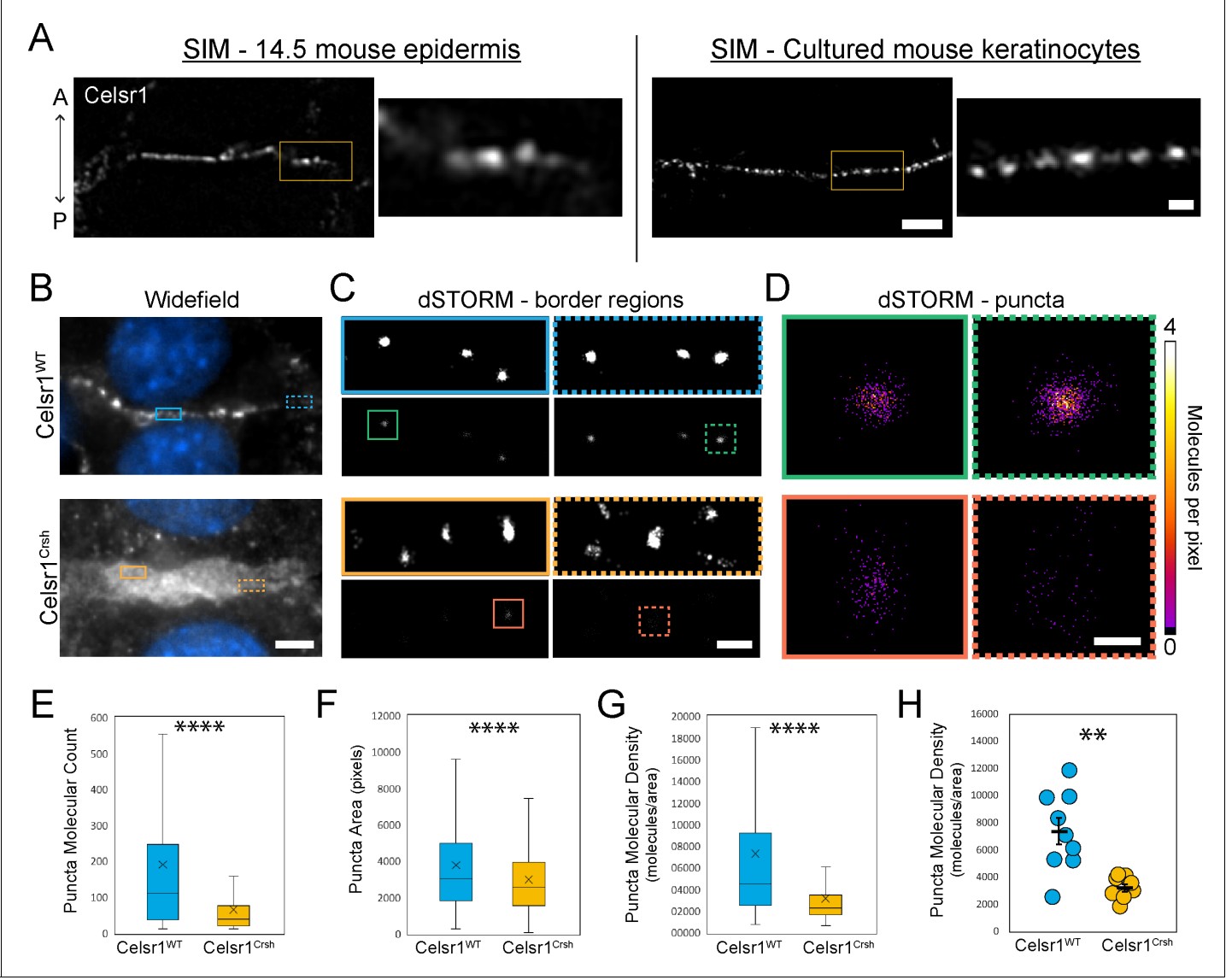

**Figure 4.** Celsr1[Crsh] disrupts the subcellular organization of Celsr1 puncta in keratinocytes as assessed by dSTORM. (**A**) Super-resolution structured illumination microscopy (SIM) images of single cell borders, stained for Celsr1, in E14.5 mouse epidermis and cultured mouse keratinocytes (cell line stably over-expressing Celsr1-mNeonGreen switched to 1.5 mM calcium for 8 hr). Images rotated to display border with horizontal orientation, and thus the epidermis image is oriented with a vertical A-P axis. Scale bar, 2 μm (inset 0.5 μm). (**B**) Widefield images of cell–cell border between Celsr1[WT] or Celsr1[Crsh] transfected cells. Scale bar, 5 μm. (**C**) dSTORM reconstruction with Gaussian (top panels) or molecular density (bottom panels) rendering of border regions highlighted in panel **A**. Scale bar, 500 nm. (**D**) dSTORM reconstruction with molecular density rendering of individual puncta highlighted in **B**. Scale bar, 100 nm. (**E–G**) Quantification of molecule count (**E**), area in pixels (**F**), and density (**G**) per puncta. Outliers not displayed. Box displays first and third quartile with mean represented by 'x'. Celsr1[WT], n = 268; Celsr1[Crsh], n = 782; two-tailed unpaired $t$-test, ****p<0.0001. (**H**) Scatter-plot of average puncta density per cell border. Black overlay represents mean ± standard error. Celsr1[WT], n = 9; Celsr1[Crsh], n = 8. **p=0.0029.

The online version of this article includes the following source data and figure supplement(s) for figure 4:

**Source data 1.** Data accompanying *Figure 4*.

**Figure supplement 1.** Celsr1 distribution in epidermis and cultured keratinocytes is punctate.

**Figure supplement 2.** Celsr1[Crsh] disrupts the subcellular organization of Celsr1 in keratinocytes.

*Figure 4—figure supplement 1*). To test how *Celsr1[Crsh]* affects the nanoscale organization of Celsr1-containing puncta, we performed super-resolution direct stochastic optical reconstruction microscopy (dSTORM), which provides ~20 nm resolution (*Rust et al., 2006*). In keratinocytes, Celsr1[WT]-GFP localized into discrete puncta that were relatively uniform in size and evenly spaced

along junctions. Despite appearing largely diffuse at cell–cell borders by widefield microscopy, Celsr1$^{Crsh}$-GFP localized into discernable puncta by dSTORM, but these puncta were distributed across a much broader junctional interface and appeared highly disorganized and less compact compared to Celsr1$^{WT}$ (*Figure 4B–D*, *Figure 4—figure supplement 2A and B*).

To characterize the molecular organization of Celsr1 puncta, individual puncta were segmented and those meeting a molecule threshold ($\geq$10) were analyzed further (*Figure 4—figure supplement 2C*). The number of pixels that failed to meet this qualifying puncta criteria was substantially higher for Celsr1$^{Crsh}$-GFP borders (*Figure 4—figure supplement 2D*), consistent with its diffuse and less clustered localization. A spatial cluster analysis, or Ripley's *K*-function (*Ripley, 1977*), confirmed that Celsr1$^{Crsh}$ does cluster beyond that expected for a random distribution, but to a lesser degree than Celsr1$^{WT}$ (*Figure 4—figure supplement 2E*). Though similar in area, we found that Celsr1$^{Crsh}$ puncta contain fewer molecules than Celsr1$^{WT}$, resulting in puncta with a dramatic reduction in molecular density (*Figure 4E–H*, *Figure 4—figure supplement 2F–H*). Together, these findings indicate that the *Celsr1$^{Crsh}$* mutation disrupts the ability of Celsr1 to laterally organize into dense, clustered assemblies.

## The *Celsr1$^{Crsh}$* mutation increases Celsr1 mobility at cell junctions

*Cis*-interactions stabilize classical cadherins at cell junctions and limit their mobility within the membrane (*Harrison et al., 2011*; *Erami et al., 2015*). The diffuse cell border distribution and inability to cluster into dense membrane domains suggested that *Celsr1$^{Crsh}$* may reduce Celsr1 stability. To measure the mobility of Celsr1 at intercellular junctions, we performed fluorescence recovery after photobleaching (FRAP) in keratinocytes expressing three Celsr1 variants – WT, Crsh, and deltaN ($\Delta$N), a deletion that removes most of the extracellular domain. Regions along cell borders between neighboring Celsr1-expressing cells were photobleached and imaged to monitor fluorescence recovery. Celsr1$^{WT}$-GFP showed little recovery into the bleached region and only a small fraction of the junctional protein was mobile, indicating Celsr1 is highly stable at junctions (*Figure 5*, *Figure 5—figure supplement 1*). Removing the extracellular domain (Celsr1$^{\Delta N}$), which prevents adhesion in trans, resulted in a dramatic increase (5.5-fold) in mobility at cell borders, demonstrating junctional stability requires intercellular interactions. The rate and total amount of fluorescence recovery for Celsr1$^{Crsh}$-GFP were intermediate compared to Celsr1$^{WT}$ and Celsr1$^{\Delta N}$ variants (3.7-fold increase relative to WT). Additionally, we observed that Celsr1$^{Crsh}$ mobility at cell borders was similar to both its mobility and that of Celsr1$^{WT}$ at non-junctional regions or 'free edges' (*Figure 5—figure supplement 2*), with only junctional Celsr1$^{WT}$ displaying substantial stability. These results suggest that the *Celsr1$^{Crsh}$* mutation lies within a region of Celsr1 that further stabilizes Celsr1 once Celsr1 is already engaged in adhesive interactions between neighboring cells. Overall, given Celsr1$^{Crsh}$ still mediates adhesion in trans, its higher mobility and diffuse distribution at cell junctions relative to Celsr1$^{WT}$ suggest that the mutation may impair lateral *cis*-interactions that function to stabilize Celsr1 adhesion.

## Induced *cis*-dimerization rescues adhesion defects caused by *Celsr1$^{Crsh}$*

Our data thus far support the hypothesis that the *Celsr1$^{Crsh}$* mutation lies within a putative *cis*-interaction domain. If correct, forcing *cis*-dimerization of the Celsr1$^{Crsh}$ mutant protein should rescue its junctional localization. To model lateral cadherin clustering, we fused the FKBP dimerization domain to the cytoplasmic tail of Celsr1$^{Crsh}$ (*Figure 6A*). Addition of a cell-permeable, small molecule dimerizer agent rapidly induces FKBP dimerization, driving lateral clustering of the tagged-cadherin (*Yap et al., 1997*; *Chen et al., 2012*; *Cadwell et al., 2016*). Keratinocytes were transfected with Celsr1$^{Crsh}$-FKBP-HA and treated with dimerizer or ethanol as a control. Dimerized Celsr1$^{Crsh}$-FKBP-HA became enriched at cell–cell borders (twofold increase) and clustered into discrete puncta, whereas the protein remained diffusely localized in ethanol-treated controls (*Figure 6B and C*, *Figure 6—figure supplement 1A*). Puncta were not readily observed outside of junctional contacts upon dimerization (refer to non-junctional regions; *Figure 6*, *Figure 6—figure supplement 1*). In agreement with non-junctional Celsr1 displaying increased mobility (*Figure 5—figure supplement 2*), these results suggest that *trans*-interactions between adjacent cells may be a prerequisite for dimerization to stabilize Celsr1 into discernable puncta.

To test whether induced dimerization could also rescue Celsr1$^{Crsh}$ interactions with the WT protein in trans, we performed a cell mixing assay (*Figure 6D*). Keratinocytes were transfected with

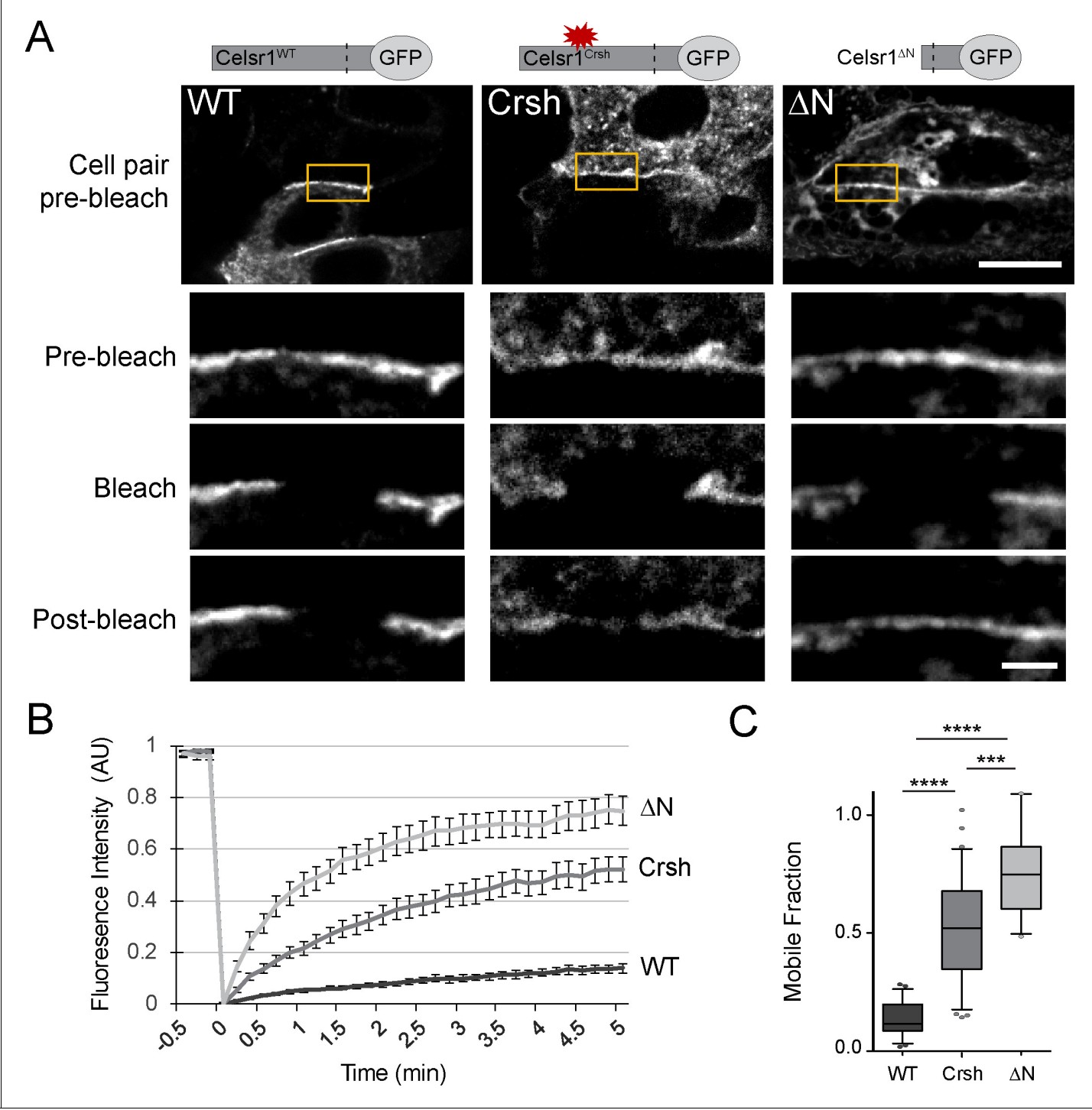

**Figure 5.** Celsr1[Crsh] displays increased mobility. (**A**) Representative images of FRAP performed on cultured keratinocytes transfected with the Celsr1-GFP variants indicated. Yellow box highlights regions (~3–4 µm in length) along cell borders that were photobleached. Images were acquired every 10 s for 5 min. Images rotated to display all borders horizontally with magnifications of borders below for the time points indicated. Pre-bleach and post-bleach images shown are at −10 s and 5 min respectively. Scale bars, 10 µm; insets, 2 µm. (**B**) Normalized FRAP recovery curves. AU, arbitrary units. (**C**) Mobile fraction quantification (normalized to 1, box and whisker plot, box displaying first quartile, mean, and third quartile). Mean ± standard error: WT (13.6 ± 1.61%, n = 23), Crsh (50.7 ± 3.9%, n = 33), and ΔN (74.7 ± 4.2%, n = 18) border regions. Two-tailed, unpaired *t*-tests with Welch's correction: WT vs Crsh, ***p=0.0003; WT vs ΔN, ****p<0.0001; Crsh vs ΔN, ****p<0.0001. WT, wild type; Crsh, Celsr1 variant harboring *Crsh* mutation; ΔN, Celsr1 variant lacking the extracellular domain.

The online version of this article includes the following source data and figure supplement(s) for figure 5:

*Figure 5 continued on next page*

*Figure 5 continued*

**Source data 1.** Data accompanying *Figure 5*.
**Figure supplement 1.** Celsr1$^{Crsh}$ and Celsr1$^{\Delta N}$ display altered mobility by FRAP.
**Figure supplement 2.** FRAP analysis of Celsr1 mobility at cell–cell borders vs cell-free edges.

either Celsr1$^{WT}$-GFP or Celsr1$^{Crsh}$-FKBP-HA prior to mixing and junction formation. In ethanol-treated controls, neither the WT nor the Celsr1$^{Crsh}$ variant became enriched at the junctional interfaces between Celsr1$^{Crsh}$ (bottom cell of pair) and Celsr1$^{WT}$ (top cell of pair) expressing cells, consistent with the inability of the two variants to form mixed cell aggregates in K562 cells (*Figure 2*). Strikingly, in the presence of dimerizer, Celsr1$^{WT}$-GFP and Celsr1$^{Crsh}$-FKBP-HA became concentrated at cell borders, colocalizing in puncta that were similar in size and organization to cells expressing WT Celsr1 alone (*Figure 4—figure supplement 1*, *Figure 6D*, *Figure 6—figure supplement 1C*). Together, these data demonstrate that forced *cis*-dimerization rescues the adhesion defects caused by Celsr1$^{Crsh}$ in keratinocytes, supportive of the hypothesis that the *Celsr1$^{Crsh}$* mutation may selectively interfere with the ability of Celsr1 to mediate lateral, *cis*-interactions.

## Celsr1$^{Crsh}$ physically interacts with PCP proteins Frizzled6 and Vangl2

We next sought to investigate the mechanism by which Celsr1 *cis*-interactions promote PCP asymmetry. Celsr1, and its *Drosophila* homologue Fmi, are known to physically associate with core PCP components Fz6 and Vangl2, and are required for their recruitment to cell junctions (*Bastock et al., 2003*; *Chen et al., 2008*; *Devenport and Fuchs, 2008*; *Strutt and Strutt, 2008*; *Struhl et al., 2012*). One possible explanation for the loss of asymmetry in *Celsr1$^{Crsh/Crsh}$* mutants is that the *Celsr1$^{Crsh}$* mutation simply abolishes the physical interaction between Celsr1 and Fz6 and/or Vangl2. However, three key pieces of evidence argue against this possibility. First, examination of Fz6 and Vangl2 localization in the epidermis in vivo showed that although their asymmetric localization was severely disrupted in *Celsr1$^{Crsh/Crsh}$* embryos, Fz6 and Vangl2 still localized to cell junctions (*Figure 7A*). Second, similar to Celsr1$^{Crsh}$ co-expression with Vangl2 (*Devenport and Fuchs, 2008*), when co-expressed with Fz6-tdTomato in cultured keratinocytes, Celsr1$^{Crsh}$-GFP was still able to drive the redistribution of Fz6 from the cytoplasm to cell contacts (*Figure 7—figure supplement 1*). Third, Celsr1$^{Crsh}$-GFP was able to co-immunoprecipitate both Fz6-tdTomato and tdTomato-Vangl2 from keratinocyte extracts (*Figure 7B and C*), indicating that the Celsr1$^{Crsh}$ variant retains the ability to physically interact with both core PCP proteins. Thus, we find that the loss of asymmetry in *Celsr1$^{Crsh/Crsh}$* mutants cannot be attributed to a lack of physical interactions between Celsr1 and Fz6 or Vangl2.

## Celsr1$^{Crsh}$ fails to organize Fz6 and Vangl2 into asymmetric PCP complexes in vivo

To determine the nanoscale organization of intercellular PCP complexes in vivo, we turned to super-resolution SIM. Given the ~150–200 nm size of Celsr1 puncta observed by dSTORM (*Figure 4*), we reasoned the ~100 nm resolution of SIM (*Galbraith and Galbraith, 2011*) would be capable of resolving individual PCP puncta, and perhaps even asymmetric Fz6 and Vangl2 localization. SIM was performed on whole mount preparations of E14.5 dorsal skin labeled with Fz6, Vangl2, and Celsr1 antibodies. This stage of embryonic development was carefully selected because the epidermis is still thin and mostly unstratified, allowing for optimal proximity to the coverslip for imaging, yet far enough along developmentally that PCP proteins are adequately polarized (*Aw et al., 2016*).

We began by analyzing Fz6 and Vangl2 organization along vertical borders (anterior–posterior junctions), where PCP proteins are enriched, and found that similar to Celsr1, Fz6 and Vangl2 colocalized in punctate assemblies. We often observed asymmetry, or separation of the two proteins across individual junctions with Fz6 toward the left, or anterior of the junction, and Vangl2 to the right, or posterior of the junction (*Figure 8A–C*). We refer to this distribution as *cross-junctional asymmetry*. Further, borders often contained multiple asymmetric puncta with Fz6 and Vangl2 partitioned in the same orientation. Although SIM was able to resolve Fz6-Vangl2 asymmetry across junctions, the separation was subtle and a large fraction of their red and green fluorescent signals overlapped, appearing as yellow, colocalized signal. Nevertheless, we reasoned the increased

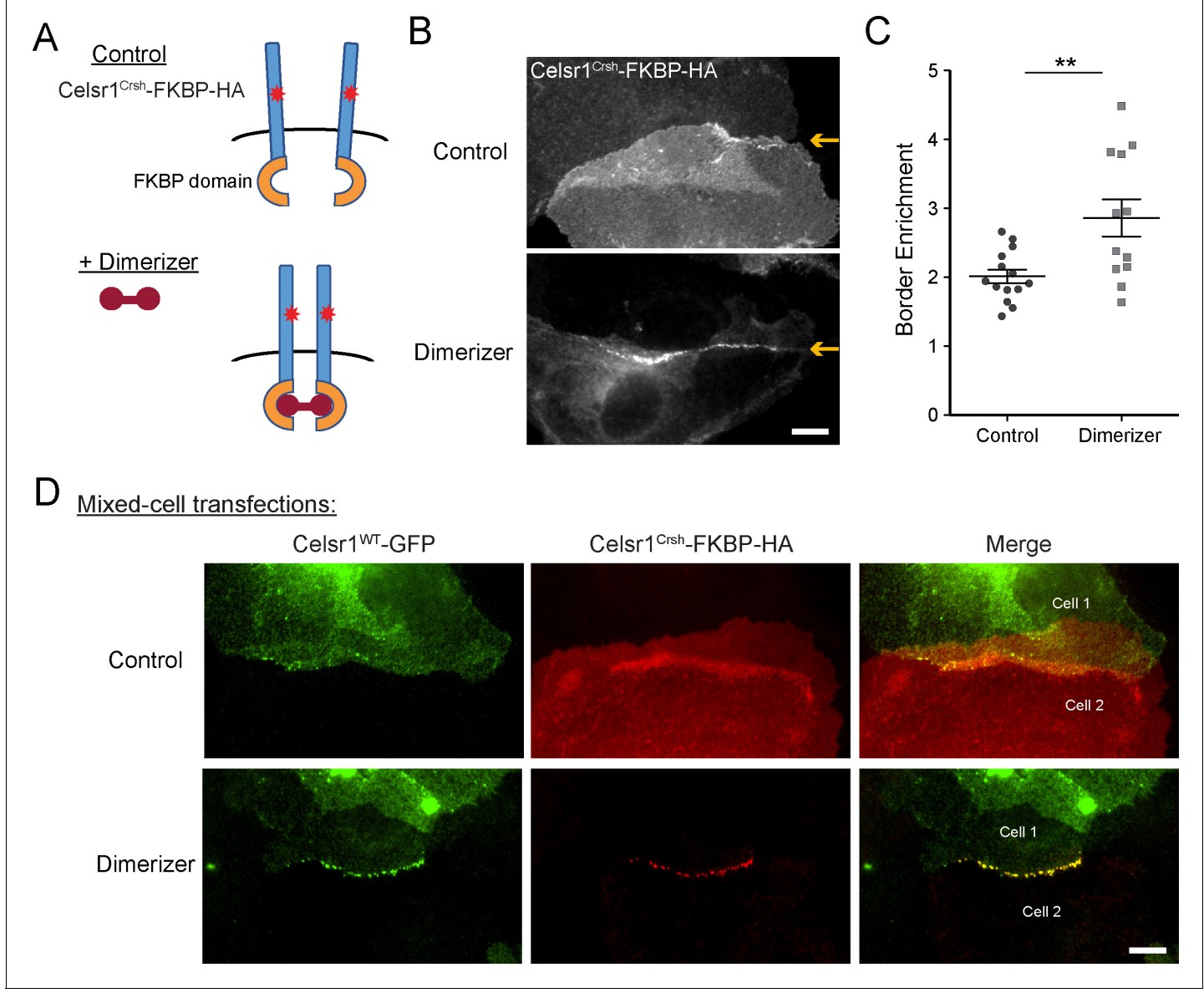

**Figure 6.** Induced *cis*-dimerization of Celsr1Crsh rescues junctional enrichment and *trans*-interactions with Celsr1WT. (**A**) Illustration of FKBP-dimer system. Protein of interest is tagged with FKBP domain. Cell-permeable agent B/B homodimerizer (dimerizer) forces FKBP domains together. (**B**) Cultured keratinocytes were transfected with Celsr1Crsh-FKBP-HA, switched to 1.5 mM calcium for 24 hr, and treated with ethanol (control) or dimerizer for 2 hr prior to the end of the calcium switch to induce dimerization of the FKBP domain fused to Celsr1Crsh. Yellow arrows highlight position of the cell–cell interface between Celsr1Crsh-FKBP-HA expressing cells. (**C**) Border enrichment quantification for Celsr1Crsh-FKBP-HA. Ethanol, n = 14; Dimerizer, n = 12. One-tailed, unpaired *t*-test, \*\*p=0.0057. (**D**) Cell mixing assay where cells were transfected separately with either Celsr1WT-GFP or Celsr1Crsh-FKBP-HA and mixed. Shown are representative examples of mixed-cell junctions between Celsr1WT-GFP (top cell in pair, green) and Celsr1Crsh-FKBP-HA (bottom cell in pair, red) treated with ethanol (control) or dimerizer. Scale bars, 10 μm.

The online version of this article includes the following source data and figure supplement(s) for figure 6:

**Source data 1.** Data accompanying *Figure 6*.
**Figure supplement 1.** Induced dimerization rescues Celsr1Crsh.

resolution of SIM would allow us to address how Celsr1Crsh alters the spatial organization of Fz6 and Vangl2.

We predicted that Fz6 and Vangl2 distributions might overlap completely in *Celsr1Crsh/Crsh* epidermis, indicating a failure of Fz6 and Vangl2 to partition to opposite sides of the junction. Alternatively, Fz6 and Vangl2 might organize into asymmetric puncta, displaying cross-junctional separation,

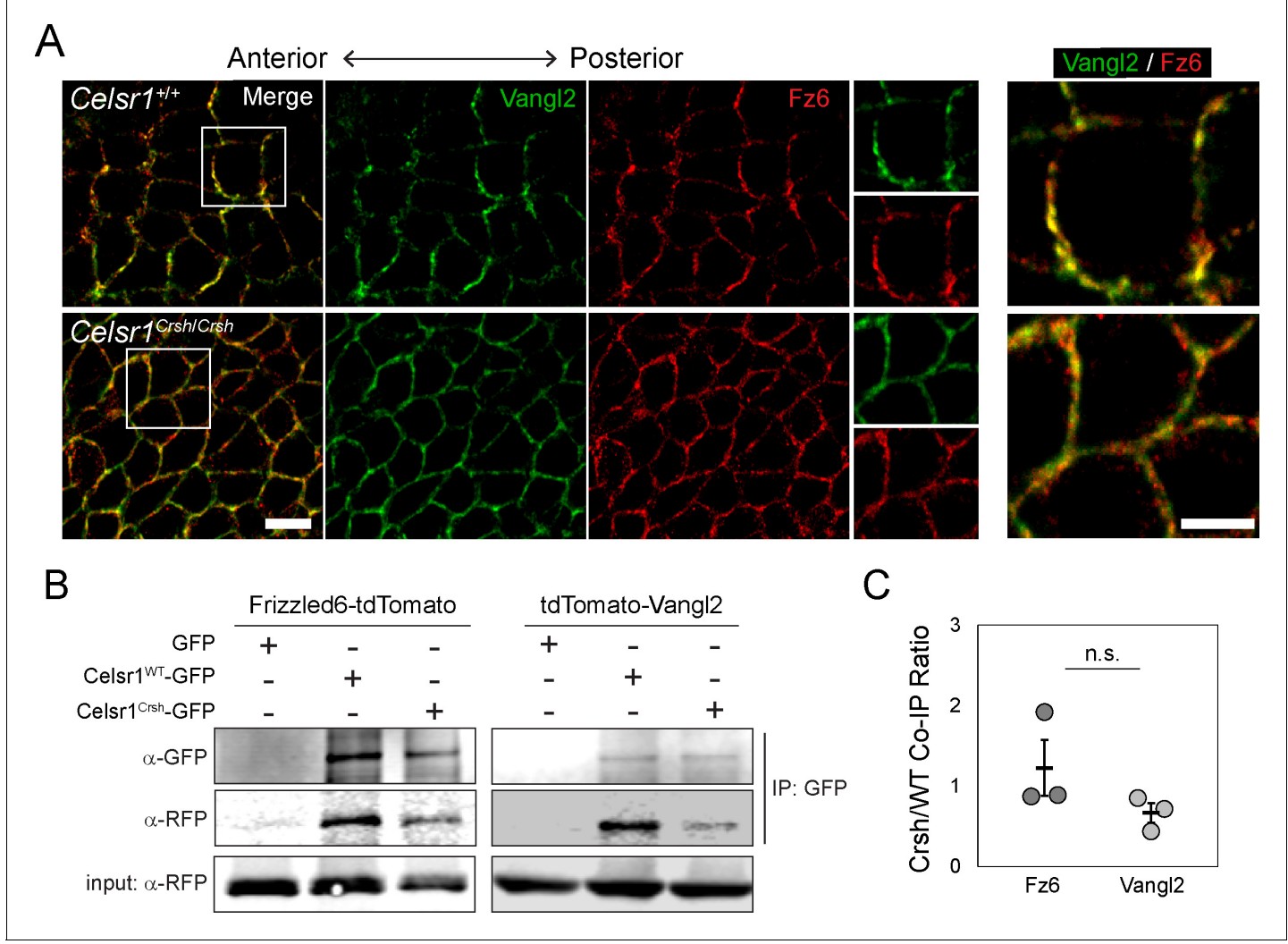

**Figure 7.** Celsr1$^{Crsh}$ interacts with planar cell polarity (PCP) proteins Frizzled6 and Vangl2. (**A**) High magnification (100×) confocal, 0.6 μm max projections of *Celsr1$^{+/+}$* or *Celsr1$^{Crsh/Crsh}$* E14.5 epidermis stained for Frizzled6 (Fz6, red) and Vangl2 (green). Scale bar, 10 μm. Right panels display two-color enlargement of white-box region on left. Scale bar, 5 μm. (**B**) Protein extracts from keratinocytes expressing Frizzled6-tdTomato or tdTomato-Vangl2 (detected with RFP antibody) with either Celsr1$^{WT}$-GFP or Celsr1$^{Crsh}$-GFP, as indicated, were immunoprecipitated with anti-GFP antibodies. (**C**) Co-immunoprecipitation (Co-IP) quantification. Immunoprecipitated Fz6 or Vangl2, normalized to Celsr1, is shown as a ratio of Crsh/WT (i.e. the normalized amount of Fz6 or Vangl2 immunoprecipitated by Celsr1$^{Crsh}$ relative to Celsr1$^{WT}$). Unpaired two-tailed *t*-test, p=0.287; n = 3.

The online version of this article includes the following source data and figure supplement(s) for figure 7:

**Source data 1.** Data accompanying *Figure 7*.

**Figure supplement 1.** Frizzled6 colocalizes with Celsr1$^{Crsh}$.

but their asymmetry would fail to align in a common orientation along cell borders. Surprisingly, we observed neither. Rather, Fz6 and Vangl2 segregated laterally along borders, into distinct, non-overlapping, and often alternating domains (*Figure 8A–C*). We refer to this distribution as *lateral-junctional separation*, which is further illustrated by plotting fluorescence intensity measurements along cell borders. Fz6 and Vangl2 signal were out of phase with each other along *Celsr1$^{Crsh/Crsh}$* borders in contrast to the cross-junctional separation and co-aligned distribution in WT (*Figure 8D*). Occasionally, cross-junctional separation was observed in *Celsr1$^{Crsh/Crsh}$* epidermis within asymmetric puncta of both correct (Fz6–Vangl2) and incorrect (Vangl2–Fz6) orientations (*Figure 8B*, images 8 and 9). The predominant phenotype, however, was lateral-junctional separation of Fz6 and Vangl2 into distinct, alternating domains. Colocalization and co-alignment of Celsr1 with either Fz6 or Vangl2 along vertical borders was similarly reduced in *Celsr1$^{Crsh/Crsh}$* epidermis (*Figure 8—figure*

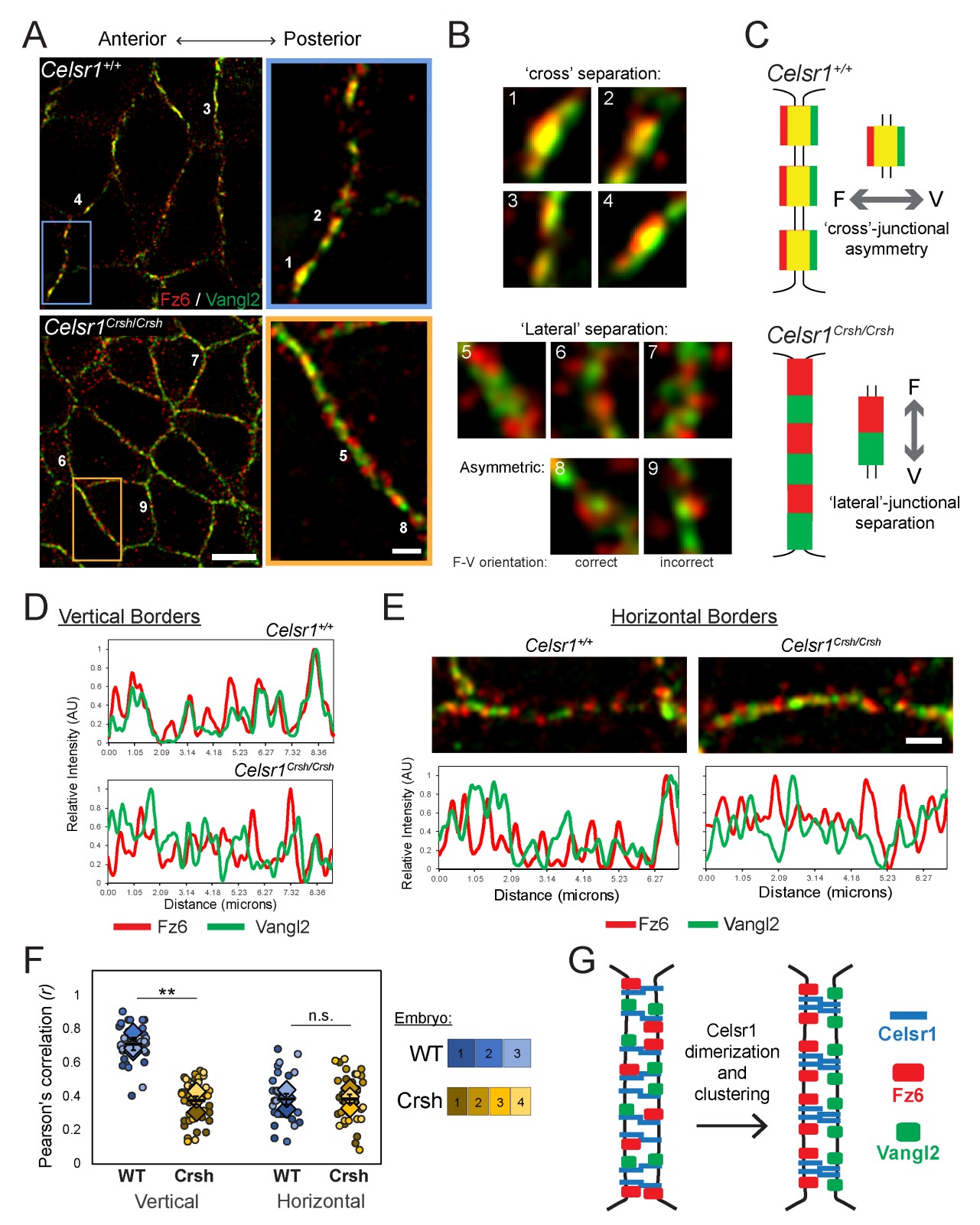

**Figure 8.** Failure to organize asymmetric planar cell polarity (PCP) complexes in *Celsr1^{Crsh/Crsh}* mutants in vivo. Whole mount preparation of E14.5 dorsal skin labeled for Fz6 and Vangl2 and imaged by super-resolution structured illumination microscopy (SIM). (**A**) Planar SIM images of the basal layer from either *Celsr1^{+/+}* or *Celsr1^{Crsh/Crsh}* embryos. Images oriented with anterior to the left. Scale bars, 5 μm (insets 1 μm). (**B**) Enlargements of individual puncta (or equivalent regions) along cell borders. (**C**) Schematic of Fz6 (F) and Vangl2 (V) distribution along cell borders that display 'cross'-

*Figure 8 continued on next page*

*Figure 8 continued*

junctional asymmetry in *Celsr1*$^{+/+}$ and 'lateral'-junctional separation in *Celsr1*$^{Crsh/Crsh}$ epidermis. (D) Fluorescence intensity measurements of lines drawn along vertical cell borders. (E) Horizontal cell borders with corresponding fluorescence intensity measurements along borders. (F) Quantification of Pearson's correlation between Fz6/Vangl2 at cell borders. Dots represent individual cell borders. Colors indicate different embryos. Vertical borders, 12 per embryo. Horizontal borders, nine per embryo. Diamonds indicate the individual embryo average. Black dash overlay represents the mean across the individual embryo average ± standard error. **p=0.0016. Data are representative of four *Celsr1*$^{+/+}$ (WT) embryos and five *Celsr1*$^{Crsh/Crsh}$ (Crsh) embryos from two litters. (G) PCP establishment model. Celsr1 dimerization and clustering through *cis*-interactions are required to organize asymmetric Fz6-Vangl2 complexes at cell junctions.

The online version of this article includes the following source data and figure supplement(s) for figure 8:

**Source data 1.** Data accompanying *Figure 8*.
**Figure supplement 1.** Colocalization analysis of Celsr1 with Fz6 or Vangl2 in vivo.
**Figure supplement 2.** Planar cell polarity (PCP) protein distribution along cell borders in vivo.

*supplement 1*; *Figure 8—figure supplement 2*). Thus, despite the ability to physically interact with both Fz6 and Vangl2 biochemically, Celsr1$^{Crsh}$ fails to properly organize asymmetric assemblies of Fz6 and Vangl2 between cells in vivo.

Though PCP proteins predominantly enrich along vertical borders, they do localize to horizontal borders in WT epidermis, albeit at lower levels (*Aw et al., 2016*). Intriguingly, we also observed lateral-junctional separation of Fz6 and Vangl2 along horizontal borders similar to their distribution in *Celsr1*$^{Crsh/Crsh}$ mutants (*Figure 8E*). To quantify the cross- and lateral-junctional separation phenotypes, we used Pearson's correlation to measure the co-occurrence of Fz6 and Vangl2 at cell borders. At vertical borders, Fz6 and Vangl2 co-occurrence was greater than 0.7 (±0.04) in WT, *Celsr1*$^{+/+}$ epidermis, and was significantly reduced in *Celsr1*$^{Crsh/Crsh}$ (0.38 ± 0.03) (*Figure 8F*). Along horizontal borders, Fz6 and Vangl2 co-occurrence was significantly lower (0.39 ± 0.03) compared to vertical borders in *Celsr1*$^{+/+}$, and was more in line with their co-localization in *Celsr1*$^{Crsh/Crsh}$ junctions (*Figure 8F*). Thus, at horizontal borders, Fz6 and Vangl2 do not organize into asymmetric co-assemblies between cells, as they do at vertical borders. Similarly, we observed lateral separation of Celsr1 from both Fz6 and Vangl2 at horizontal borders in *Celsr1*$^{Crsh/Crsh}$ epidermis (*Figure 8—figure supplement 1*). This separation of Fz6 from Celsr1 at horizontal borders occurs to a similar extent as observed in *Celsr1*$^{+/+}$ epidermis. Interestingly, Vangl2 separation from Celsr1 at horizontal borders is enhanced in *Celsr1*$^{Crsh/Crsh}$ epidermis relative to *Celsr1*$^{+/+}$ (*Figure 8—figure supplement 1*), suggesting that *Celsr1*$^{Crsh}$ may impart a greater deleterious effect on Vangl2 interactions with Celsr1. Even though Vangl2 colocalization with Celsr1 is slightly reduced at *Celsr1*$^{Crsh/Crsh}$ horizontal junctions, we collectively observe that the nanoscale organization of PCP proteins in *Celsr1*$^{Crsh/Crsh}$ mutants more closely mirrors their organization at WT horizontal borders.

Together, these data demonstrate that SIM resolves the asymmetry of endogenous Fz6 and Vangl2 across individual junctions, consistent with their predicted organization into 'asymmetric bridges' from mosaic overexpression studies. Further, these findings provide evidence to support a model (*Figure 8G*) in which *cis*-interactions that dimerize and cluster Celsr1 are required for organizing asymmetric Fz6-Vangl2 junctional assemblies.

## Discussion

Tissue organization requires collective cell interactions governed by the PCP pathway that align cell polarity across entire tissues. The asymmetric localization of PCP proteins at cell junctions, a hallmark of functional PCP, requires Celsr1 adhesion between neighboring cells. Here, we provide evidence that mammalian Celsr1 functions not only as a *trans*-adhesive homodimeric bridge but also as an organizer of intercellular Fz6 and Vangl2 asymmetry through lateral, *cis*-interactions.

Cadherin-based adhesion is known to involve both *trans*- and *cis*-interactions of the extracellular cadherin repeats (*Niessen et al., 2011*; *Brasch et al., 2012*; *Priest et al., 2017*). Although Celsr1 enrichment at keratinocyte cell interfaces was highly suggestive of adhesive activity, our aggregation experiments provide definitive evidence that Celsr1 mediates cell–cell adhesion in trans. This agrees with the aggregation ability of *Drosophila* Fmi in S2 cells (*Usui et al., 1999*) and of mammalian Celsr2 and Celsr3 in A431D cells (*Shima et al., 2007*). Further, we and others have shown that adhesion within the Fmi protein family requires the extracellular domain, specifically the cadherin repeats

(*Kimura et al., 2006*; *Shima et al., 2007*). It remains to be determined whether the Celsr1 cadherin repeats are sufficient for adhesion and which cadherin repeats engage in homophilic binding.

Using the Celsr1$^{Crsh}$ variant to investigate Celsr1 adhesive interactions in PCP, we provide several lines of evidence that Celsr1 also engages in lateral *cis*-interactions. Though, it remains to be determined if the *Celsr1$^{Crsh}$* mutation weakens *trans*-interactions, we find that Celsr1$^{Crsh}$ retains the ability to mediate *trans*-cell adhesion in aggregation assays. Cadherin mutations that disrupt *trans*-interactions abolish cell adhesion and mutations that interfere with *cis*-interactions prevent the formation of stable junctions (*Harrison et al., 2011*). This is precisely what we observe for Celsr1$^{Crsh}$. Cell adhesion in the aggregation assay is not abolished, rather we find that Celsr1$^{Crsh}$ fails to form stable junctions, both on its own or in mixed junctions with WT. Whereas Celsr1$^{WT}$ enriches at cell borders and organizes into punctate assemblies, the Celsr1$^{Crsh}$ variant displays a diffuse cell surface distribution and fails to enrich at cell–cell borders. Further, Celsr1$^{Crsh}$ exhibits increased mobility by FRAP, but is more stable than a variant incapable of mediating adhesion in trans. Notably, many of these features phenocopy those of an E-cadherin *cis*-mutant (*Harrison et al., 2011*). Finally, induced *cis*-dimerization of Celsr1$^{Crsh}$ rescues both its border enrichment and interactions with Celsr1$^{WT}$, providing compelling evidence that Celsr1$^{Crsh}$ interferes with extracellular *cis*-interactions. The Celsr1 *cis*-binding interface could involve EC7-8, near where the *Celsr1$^{Crsh}$* mutation lies. Alternatively, many cadherin superfamily proteins contain membrane adjacent domains (MADs) that mediate weak *cis*-homodimerization (*De-la-Torre et al., 2018*). Celsr1 contains a predicted MAD just C-terminal to EC9, thus it is possible that the *Celsr1$^{Crsh}$* mutation hinders MAD-dependent *cis*-interactions.

Cadherins organize into relatively stable, nanoscale clusters (*Yap et al., 2015*), a feature we observe to be shared by Celsr1 in both mouse epidermis and keratinocytes. Further, PCP proteins also organize into stable punctate assemblies that are highly clustered, uniformly sized, and regularly spaced along cell junctions (*Strutt et al., 2011*; *Aw et al., 2016*; *Strutt et al., 2016b*; *Warrington et al., 2017*). In *Drosophila*, these puncta are characterized by increased fluorescence intensity and low turnover compared to non-punctate regions (*Strutt et al., 2011*; *Strutt et al., 2016b*; *Warrington et al., 2017*). Our data implicate the EC7-8 region of Celsr1 in this stable cluster formation. Using dSTORM, we observe that Celsr1 organizes into dense, evenly spaced puncta at cell borders, whereas Celsr1$^{Crsh}$ mutant puncta are disorganized with reduced molecular density. Celsr1 turnover is also relatively slow, with a mobile fraction only 60% that of E-cadherin (*Aw et al., 2016*), and the *Celsr1$^{Crsh}$* mutation increases Celsr1 mobility 3.7-fold. Thus, enrichment of PCP proteins into stable punctate assemblies is a conserved feature of PCP organization driven in part by Celsr1-mediated lateral *cis*-clustering.

Importantly, our data also provide new insights into the formation of asymmetric, junctional PCP complexes. Using SIM, we observe separation of Fz6 and Vangl2 across vertically oriented junctions, providing direct evidence that endogenous Fz6 and Vangl2 partition to opposite sides of the cell. By contrast, Fz6 and Vangl2 segregate laterally along junctions in *Celsr1$^{Crsh/Crsh}$* epidermis, suggesting that Celsr1 *cis*-clustering is a prerequisite for intercellular Fz6-Celsr1 and Vangl2-Celsr1 complexes to stably engage. Intriguingly, Fz6 and Vangl2 also separate laterally along horizontal, non-PCP borders suggesting that lateral-junctional separation may represent a distinct phase of PCP establishment or an alternative steady-state of PCP. Although we cannot resolve whether laterally separated Fz6 and Vangl2 domains represent symmetric, homotypic PCP complexes, or hemi-PCP complexes that are unbound intercellularly, such non-asymmetric complexes are thought to be unstable and removed through endocytic turnover (*Strutt and Strutt, 2008*). The observation that PCP organization at horizontal borders resembles that of a potential *cis*-clustering mutant suggests a model where global cues that direct PCP asymmetry might promote Celsr1 lateral clustering, ensuring stable PCP complexes form preferably along vertical edges.

*Drosophila* Fmi has been shown to display differential preferences for Fz and Vang (*Chen et al., 2008*; *Strutt and Strutt, 2008*; *Struhl et al., 2012*), leading to the suggestion that the Fmi–Fmi *trans*-homodimer could have an asymmetric conformation (*Chen et al., 2008*). Thus, Celsr1 *cis*-interactions may in fact provide such a conformational change that serves as an allosteric switch leading to preferential Fz6 or Vangl2 interactions. This in turn could nucleate and propagate asymmetry via clustered, stable PCP domains. Feedback interactions involving Fz6, Vangl2, and/or cytoplasmic PCP proteins could then amplify asymmetry through additional Celsr1 clustering and stabilization.

The importance of investigating how Celsr1 adhesion contributes to junctional PCP asymmetry and the utility of the *Celsr1$^{Crsh}$* mutation are underscored by the presence of human disease-

associated mutations in Celsr1, such as those identified in patients with neural tube and congenital heart defects (*Allache et al., 2012*; *Qiao et al., 2016*). Several of these human Celsr1 mutations have been mapped to the cadherin repeats, including the region near the *Celsr1^Crsh* mutation in mouse Celsr1. Of particular interest are variants P870L and V1008L, as they may hinder Celsr1 dimerization and PCP organization, thus contributing to the disease phenotype. V1008L maps near the *Celsr1^Crsh* region of Celsr1 and P870L maps to EC6, a membrane-proximal cadherin repeat not far from EC8 where *Celsr1^Crsh* is localized. Interestingly, a zebrafish model of P870L induced neural tube defects, but the underlying molecular interactions and membrane dynamics of Celsr1^P870L need to be investigated further. Future studies deciphering whether these and other human mutations differentially disrupt Celsr1 adhesion and PCP asymmetry will further aid in identifying the mechanisms of Celsr1 adhesion.

Together with the data presented here, we propose a model in which the *Celsr1^Crsh* mutation lies within a *cis*-interacting region of Celsr1 that functions to stabilize and reinforce adhesion. Lateral *cis*-interactions would serve as a novel mechanism to regulate Celsr1 adhesion, stability, and possible conformational changes that are required for the asymmetric organization of core PCP proteins within the junctional complex. Additional studies that define the precise *cis*- and *trans*-binding interfaces are essential to support or challenge this model of Celsr1 adhesion and function. Elucidating the specific Celsr1 adhesive interactions that contribute to PCP complex organization will further illuminate how PCP modulates tissue polarity during development and disease.

# Materials and methods

## Key resources table

| Reagent type (species) or resource | Designation | Source or reference | Identifiers | Additional information |
|---|---|---|---|---|
| Strain, strain background (*M. musculus*) | *Celsr1^Crsh/+* (mixed background) | Elaine Fuchs, Jen Murdoch (*Curtin et al., 2003*) | MGI: 2668337 | |
| Cell line (*Homo-sapiens*) | K-562 | ATCC | CCL-243 | |
| Cell line (*M. musculus*) | Celsr1-mNeonGreen (mouse CD1 keratinocytes) | *Heck and Devenport, 2017* | | |
| Antibody | Anti-Celsr1 (Guinea pig polyclonal) | *Devenport and Fuchs, 2008* | | 1:1000 |
| Antibody | Anti-Frizzled6 (Goat polyclonal) | R and D Systems | Cat #AF1526 | 1:400 |
| Antibody | Anti-Vangl2, clone 2G4 (Rat monoclonal) | Millipore | Cat #MABN750 | 1:100 |
| Antibody | Anti-GFP (Chicken polyclonal) | Abcam | Cat #ab13970 | 1:2000 (IF), 1:2500 (WB) |
| Antibody | Anti-RFP, also recognizes mCherry (Rabbit polyclonal) | Rockland Inc | Cat #600-401-379 | 1:200 (IF), 1:1000 (WB) |
| Antibody | Anti-HA, clone 3F10 (Rat monoclonal) | Roche | Cat #11867423001 | 1:500 |
| Antibody | Anti-E-cadherin, clone DECMA-1 (Rat monoclonal) | Thermo Pierce | Cat #MA1-251-60 | 1:1000 |
| Antibody | Anti-Integrin β1 (Mouse monoclonal) | Millipore | Cat #MAB1965 | 1:1000 |
| Antibody | Anti-Guinea Pig, Alexa Fluor 488 (Donkey) | Jackson ImmunoResearch | Cat #706-545-148; | 1:2000 |

*Continued on next page*

*Continued*

| Reagent type (species) or resource | Designation | Source or reference | Identifiers | Additional information |
|---|---|---|---|---|
| Antibody | Anti-Guinea Pig, Alexa Fluor 647 (Donkey) | Jackson ImmunoResearch | Cat #706-605-148 | 1:2000 |
| Antibody | Anti-Guinea Pig, Alexa Fluor 555 (Goat) | Invitrogen | Cat #A-21435 | 1:2000 |
| Antibody | Anti-Chicken, Alexa Fluor 488 (Goat) | Invitrogen | Cat #A-11039 | 1:2000, 1:500 (dSTORM) |
| Antibody | Anti-Rabbit, Alexa Fluor 555 (Donkey) | Invitrogen | Cat #A-31572 | 1:2000 |
| Antibody | Anti-Rat, Alexa Fluor 647 (Donkey) | Jackson ImmunoResearch | Cat #712-605-153 | 1:2000 |
| Antibody | Anti-Rat, Alexa Fluor 555 (Goat) | Invitrogen | Cat #A-21434 | 1:2000 |
| Antibody | Anti-Goat, Alexa Fluor 555 (Donkey) | Invitrogen | Cat #A-21432 | 1:2000 |
| Antibody | Anti-Chicken IRdye 680RD (Donkey) | LI-COR | Cat #925–68075 | 1:10,000 |
| Antibody | Anti-Mouse IRDye 800CW (Goat) | LI-COR | Cat #926–32210 | 1:10,000 |
| Antibody | Anti-Rabbit IRDye 800CW (Goat) | LI-COR | Cat #926–3221 | 1:10,000 |
| Chemical compound, drug | B/B Homodimerizer | TakaraBio, Clontech | Cat #635059 | 1:1000 |
| Chemical compound, drug | Catalase | Sigma | Cat #C40 | |
| Chemical compound, drug | Glucose oxidase | Sigma | Cat #G2133 | |
| Chemical compound, drug | G418 | Fisher | Cat #50-841-720 | |
| Transfection reagent | Effectene | QIAGEN | Cat #301427 | |
| Transfection reagent | SuperFect | QIAGEN | Cat #301305 | |
| Other | EX-Link Sulfo-NHS-SS-Biotin | Thermo Scientific | Cat #21331 | |
| Other | Pierce Streptavidin Agarose | Thermo Scientific | Cat #20347 | |
| Other | Protease inhibitor cocktail | Roche | Cat #11836153001 | |
| Recombinant DNA reagent | pEGFPN1 (for GFP only control) (plasmid) | Clontech | | |
| Recombinant DNA reagent | pEGFPN1-Celsr1-GFP (plasmid) | *Devenport and Fuchs, 2008* | | |
| Recombinant DNA reagent | pEGFPN1-Celsr1-mCherry (plasmid) | This paper | | |

*Continued on next page*

*Continued*

| Reagent type (species) or resource | Designation | Source or reference | Identifiers | Additional information |
|---|---|---|---|---|
| Recombinant DNA reagent | pEGFPN1-Celsr1-Crsh-GFP (plasmid) | *Devenport and Fuchs, 2008* | | |
| Recombinant DNA reagent | pEGFPN1-Celsr1-Crsh-mCherry (plasmid) | This paper | | |
| Recombinant DNA reagent | pEGFPN1-Celsr1-Crsh-FKBP-HA (plasmid) | This paper | | |
| Recombinant DNA reagent | pEGFPN1-Celsr1-dN-GFP (plasmid) | This paper | | |
| Recombinant DNA reagent | pEGFPN1-EcadECD+TM-Celsr1CT-mCherry (plasmid) | This paper | | |
| Recombinant DNA reagent | K14-Fzd6-mCherry (plasmid) | *Devenport and Fuchs, 2008* | | |
| Recombinant DNA reagent | K14-Fzd6-tdTomato (plasmid) | *Heck and Devenport, 2017* | | |
| Recombinant DNA reagent | K14-tdTomato-Vangl2 (plasmid) | This paper | | |
| Software, algorithm | Coloc2 | Fiji (ImageJ) | https://imagej.net/Coloc_2 | |
| Software, algorithm | Matlab | MathWorks | https://www.mathworks.com/products/matlab.html | |
| Software, algorithm | R | R Project | https://www.r-project.org/ | |
| Software, algorithm | spatstat | *Baddeley et al., 2016* | http://www.crcpress.com/Spatial-Point-Patterns-Methodology-and-Applications-with-R/Baddeley-Rubak-Turner/9781482210200/. | |
| Software, algorithm | Prism | GraphPad | https://www.graphpad.com/scientific-software/prism/ | |
| Software, algorithm | Tissue Analyzer | *Aigouy et al., 2010* | https://grr.gred-clermont.fr/labmirouse/software/WebPA/index.html | |

## Mouse lines and breeding

Mice were housed in accordance with the Guide for the Care and Use of Laboratory Animals in an AAALAC-accredited facility. Animal maintenance and husbandry were in full compliance with the laboratory Animal Welfare Act. All procedures involving animals were approved by Princeton University's Institutional Animal Care and Use Committee (IACUC). Mixed background cryopreserved $Celsr1^{Crsh/+}$ were a kind gift from Elaine Fuchs (Rockefeller University) with permission from Jennifer Murdoch (Royal Holloway University of London, UK), and with rederivation done through the Rutgers Cancer Institute of New Jersey. Experiments were performed on E14.5–15.5 embryos (male and female) from matings of heterozygous mice.

## Cells and cell lines

Primary mouse keratinocytes, passage 17–21, derived from *Mus musculus* CD-1 (Charles River, Strain #022) P0 dorsal epidermis in the Devenport Lab were cultured in E-medium supplemented with 15% fetal bovine serum (FBS) and 0.05 mM $Ca^{2+}$, at 37°C with 5% $CO_2$, and were free of mycoplasma. For immunofluorescence experiments, cells were cultured on #1.5 glass coverslips coated with fibronectin. Keratinocytes were switch to 1.5 mM $Ca^{2+}$ for the durations indicated to induce junction assembly. Effectene transfection reagent (QIAGEN) was used for transfection prior to processing for

immunofluorescence as described below. K-562 cells (CCL-243, ATCC, Manassas, VA) were cultured in Iscove's Modified Dulbecco's Medium (IMDM; ATCC, Cat #30–2005 or Fisher, Cat #SH30228) supplemented with 10% FBS at 37°C with 5% $CO_2$. Stable K-562 cell lines expressing various Celsr1 variants and controls as indicated were generated by FACS sorting post-transfection with SuperFect transfection reagent (QIAGEN) and selection with G418.

## Molecular cloning and constructs

Celsr1[Crsh]-FKBP-HA was generated by replacing EGFP in Celsr1[Crsh]-GFP (*Devenport and Fuchs, 2008*) with HA-tagged FKBP (pC4-Fv1E; ARGENT Regulated Homodimerization Kit, Ariad) digested with *HindIII* and *NotI* sites. K14-tdTomato-Vangl2 was constructed by replacing the mCherry cassette (K14-mCherry-Vangl2 *Devenport and Fuchs, 2008*) with tdTomato via InFusion cloning. For stable K-562 cell lines, Celsr1[ΔN]-GFP (*Devenport et al., 2011*) was subcloned into the pEGFPN1 vector (Clontech) using *EcoRI* and *Not1* sites, WT Celsr1-mCherry was generated by replacing the EGFP cassette of Celsr1[WT]-GFP (pEGFPN1-Celsr1 *Devenport and Fuchs, 2008*) with mCherry using *HindIII* and *NotI* sites, and mCherry was subcloned in place of GFP for Celsr1[Crsh]-GFP (*Devenport and Fuchs, 2008*) and E-Cad-Celsr1[CT]-GFP (*Devenport et al., 2011*) to generate mCherry-tagged variants.

## Co-immunoprecipitation and western blot analysis

Keratinocytes were co-transfected with Celsr1[WT]-GFP, Celsr1[Crsh]-GFP, or GFP and either tdTomato-Vangl2 or Fz6-tdTomato. Cells were lysed for 1 hr on ice in cadherin extraction buffer (50 mM Tris pH 7.4, 150 mM NaCl, 5 mM $CaCl_2$, 56 mM $MgCl_2$, 1% NP40, 1% Triton X) following 48 hr post-transfection with an 8 hr calcium shift. Celsr1[WT]-GFP, Celsr1[Crsh]-GFP, or GFP immunoaffinity purification was performed using GFP-Trap_MA GFP antibody-coupled magnetic beads (Chromotek) overnight at 4°C. Proteins were resolved on a 7.5% SDS gel and transferred to a nitrocellulose membrane (Bio-Rad). Standard protocols were performed for western blotting using primary antibodies against RFP (rabbit, Rockland Inc, 1:1000) and GFP (chicken, Abcam, 1:2500), and IRDye680 and IRDye800 secondary antibodies (LI-COR, 1:10,000). Immunofluorescence detection of the bands was performed using the LI-COR Odyssey CLx imaging system.

## Surface biotinylation

K-562 cells expressing either Celsr1[WT]-mCherry or Celsr1[Crsh]-GFP were seeded in a 6-well tissue plate and biotinylated with PBS+ (containing calcium and magnesium) containing 0.5 mg/mL EX-Link Sulfo-NHS-SS-Biotin at 37°C for 30 min. The following was carried out at 4°C: unbound biotin was quenched with cold PBS+ containing 50 mM Tris (pH 8.0); following cold PBS+ washes, cells were lysed in RIPA buffer containing 1% Triton X-100, 0.1% SDS, 0.1% sodium deoxycholate, 10 mM Tris-HCl (pH 8.0), 140 mM NaCl, 1 mM EDTA, 0.5 mM EGTA, and protease inhibitors for 10 min; samples were subjected to centrifugation at 13,200 RPM for 10 min and biotinylated protein from the supernatant were incubated with streptavidin beads for 1 hr with rotation; beads were captured by centrifugation at 2500 × g for 1 min and washed with cold PBS+. Proteins were released from beads with the addition of sample buffer. Samples were processed for western blot analysis as described above.

## K-562 cell aggregation and processing

K-562 cell lines were cultured for a minimum of 24 hr post thawing prior to aggregation experiments. Cells were washed with PBS without calcium twice and resuspended in fresh growth medium. Cells were then placed on a Gyro mini shaker in an incubator at 37°C and 5% $CO_2$ for 1–24 hr as indicated to allow for aggregation. Cells were either imaged immediately following aggregation or fixed in 4% PFA for 10 min. Cells were rinsed with PBS and cleared in 50% glycerol for imaging. Cell aggregates were imaged by widefield or confocal microscopy.

## Immunostaining

Cultured keratinocytes on #1.5 coverslips were fixed in 4% paraformaldehyde (PFA) in PBS for 10 min at room temperature. Cells were washed three times in PBS and permeabilized in 0.1% Triton X-100 (in PBS, PBT) with subsequent PBT washes. Cells were incubated with primary antibodies (in

PBT) for 1 hr and secondary antibodies for 30 min with PBT washes in between. Coverslips were washed and mounted in ProLong Gold antifade reagent (Molecular Probes). For whole mount dorsal skin preparations, E14.5 or E15.5 embryos were dissected in PBS+ to maintain calcium-dependent epidermal cell–cell adhesions and fixed in 4% PFA (in PBS+) for 1 hr at room temperature with rocking. Dissected skins were blocked in 0.2% Triton X-100 (PBT2) containing 2.5% normal donkey serum, 1% BSA, and 1% fish gelatin for 2 hr at room temperature or overnight at 4°C. Skins were incubated with primary antibodies overnight at 4°C followed by three 30 min PBT2 washes at room temperature and secondary antibodies plus Hoechst (1:2000) labeling for 2–3 hr at room temperature or overnight at 4°C. Following additional washes in PBT2 and PBS, skins were mounted in either ProLong Gold or Glass mounting reagent (Invitrogen).

## Microscopy and image processing

Unless specified, widefield images were acquired on a Nikon Ti inverted epifluorescent microscope equipped with Plan Apo objectives (4×/0.13NA, 10×/0.45NA, 20×/0.75NA, 40×/1.3NA oil, 60×/1.4NA oil, 100×/1.4NA oil), four channel epifluorescent filters, CCD camera, and motorized stage.

### Confocal microscopy

Confocal images were acquired on either an inverted Nikon A1 or a Nikon A1R-Si confocal microscope equipped with a Nikon Eclipse Ti stand (Nikon Instruments), Galvano scanner, GaAsP detector, and a Plan Fluor 40X/1.3 NA or Plan Apo 100X/1.45NA oil immersion objective (Nikon).

### Structured illumination microscopy

SIM images were acquired in 3D-SIM mode using a Nikon N-SIM Eclipse Ti-E system (Nikon Instruments; Melville, NY) equipped with 488, 561, and 647 nm laser lines, a CFI SR HP Apochromat TIRF 100X/1.49 NA oil objective, and EMCCD camera (DU-897 X-8453, Andor; UK). Images were reconstructed using the N-SIM module of NIS-Elements software (Nikon; Melville, NY).

### Direct stochastic optical reconstruction microscopy

Cells were cultured on 8-well #1.5 glass bottom μ-slides (Cat#80827; Ibidi, Germany). Fixation and labeling protocols were adapted from Whelan and Bell as described previously (*Whelan and Bell, 2015*; *Stahley et al., 2016*). Briefly, cells were fixed with 4% PFA prepared fresh from 16% electron microscopy grade material (Electron Microscopy Sciences; Hatfield, PA). Cells were washed, blocked, and permeabilized followed by primary and secondary antibody incubations. Samples were imaged within 2 weeks and in fresh imaging buffer containing glucose oxidase, catalase, and β-mercaptoethanol for photoswitching. dSTORM images were acquired on a Nikon Ti-E with Perfect Focus System (interferometric based focus maintenance [PFS]), CFI SR HP Apochromat TIRF 100×/1.49NA oil objective, Agilent high power MLC400 70 mW 488 nm laser line, and 512 back-thinned EMCCD (iXon Ultra; Andor, UK). 30,000 frames were collected with inclined sub-critical excitation at 47 fps per image. Images were reconstructed with NIS-Elements software using a minimum height of 2500 for molecule identification and exported at 3 nm/pixel with either Gaussian or density map rendering. Corresponding widefield images were obtained on the N-STORM system just prior to dSTORM acquisition.

### Fluorescence recovery after photobleaching

CD1 mouse primary keratinocytes (passage 19–21) were cultured on #1.5 glass bottom dishes (35 mm, Cat# 81218; Ibidi, Germany) in E-media with added calcium (50 μM). Cells were co-transfected with the GFP-tagged Celsr1 variants indicated and Celsr1$^{WT}$-mCherry using Effectene transfection reagent (QIAGEN) for 48 hr. Celsr1$^{WT}$-mCherry was used to identify cell border regions of comparable morphology while the Celsr1-GFP variants were used for analysis. To induce junction formation, cells were switched to high calcium (1.5 mM) E-media for 24 hr prior to imaging. FRAP experiments were performed on a Yokogawa spinning disc (CSU-21) confocal mounted on a Nikon Ti-E with PFS, Agilent laser launch with 488 and 561 nm lasers, Hamamatsu Orca-Flash4.0 sCMOS camera, and a Plan Apo TIRF 100X/1.45 NA oil immersion objective (Nikon). Cells were maintained in 5% $CO_2$ at 37°C. Images were captured at a single z-plane, every 10 s for 30 s prior to photobleaching. For FRAP stimulation, a linear 3–4 μm region encompassing cell junction areas was used for stimulation

region of interest (ROI) in all experimental setups. Photobleaching was achieved with a Bruker X-Y mini-scanner module and 405 nm OBIS laser (100 mW, Coherent; Santa Clara, CA). Time-lapse images were captured every 10 s for a recovery period of 5 min after photobleaching. Normalized FRAP recovery curves were generated by determining the normalized intensity as $(F_t - F_{bleach}) / (F_{pre} - F_{bleach})$, where $F_t$ represents the fluorescence intensity at times points during the acquisition. Mobile fractions were calculated as: $M(f) = (F_{end} - F_{bleach}) / (F_{pre} - F_{bleach})$, where $F_{end}$ represents the average of the final three timepoints, $F_{bleach}$ represents the timepoint immediately following the bleach stimulation, and $F_{pre}$ represents the fluorescence intensity at the ROI before photobleaching.

## Image processing and analysis

ImageJ Fiji was used for image processing unless noted otherwise. Confocal and SIM images were background subtracted in ImageJ with a rolling ball radius of 100 and 50 px respectively. Figures were assembled in Adobe Illustrator CS5.1.

### Quantification of Celsr1 border enrichment

Border enrichment is defined as the ratio of Celsr1 fluorescence at cell borders relative to non-border regions. For *Figure 1F*, the ratio of mean border intensity relative to mean intensity of the cell pair using segmented ROIs was quantified. For *Figure 4*, a segmented line was drawn along the junction and non-junction regions of the cell perimeter and the ratio of junction to non-junction mean intensity ratio was quantified.

### Quantification of Celsr1 polarity

Celsr1 polarity was calculated using Tissue Analyzer as previously described (*Aigouy et al., 2010*; *Aw et al., 2016*; *Cetera et al., 2017*). Briefly, cells were segmented using Vangl2 staining as a membrane signal for images in *Figure 1*. Axis and magnitude (nematic order) of junctional polarity were calculated. MATLAB was used to plot the data on a circular histogram using the polar plot function. The magnitude of average Celsr1 polarity (Mp) was defined previously (*Aigouy et al., 2010*).

### Quantification of Celsr1 puncta imaged by dSTORM

After initial processing of raw dSTORM images in NIS-Elements, cell border regions of molecular density rendered images were further processed using custom MATLAB scripts (source codes 1–2). Images were binarized and Celsr1-containing clusters were identified using a morphological closing operation with a disk-shaped structural element and a 17-pixel radius corresponding to 45 nm. Identified clusters were labeled with IDs using the 'bwlabel' function in MATLAB with an 8-pixel connectivity. The labeled image was then multiplied with the original binarized image to label the original pixels with the associated cluster ID. Clusters with 10 or more molecules were classified as 'puncta'. Coordinates of all pixels were outputted and marked whether they were part of a puncta or not. For those classified as puncta, several measurements were taken. Molecule count is the sum of the value of pixels in the puncta of the image prior to binarization. Area of the pixels making up a puncta was defined using a convex hull generated from the puncta pixels. Density is the number of molecules in a puncta divided by the area of the convex hull for that puncta. Images of puncta outlined with the convex hull were generated to allow for manual filtering of puncta actually containing multiple puncta that were mis-identified as individual puncta. Fifteen out of 283 Celsr1$^{WT}$ (WT) and 14 out of 795 Celsr1$^{Crsh}$ (Crsh) puncta were filtered out of all further analyses and plots.

To quantify Celsr1 clustering behavior, we turned to the widely utilized Ripley's K function, a measure of spatial clustering by determining the deviation of the observed distribution from a random, non-clustered distribution (*Ripley, 1977*; *Gao et al., 2015*). Ripley's L statistic, the linear transformation of Ripley's K, was calculated on the coordinate data in a custom R script (*Source code 3*) using the 'spatstat' package (*Baddeley et al., 2016*). Coordinates were read into R and the point pattern data set was constructed for each cluster with windows as rectangles with width equal to the distance between the leftmost and rightmost point plus 75 pixels (200 nm), and height equal to the distance between the bottommost and topmost point plus 75 pixels. This 200 nm buffer was added to ensure that the Ripley's L statistic was calculated up to 200 nm for each puncta, as the sizes of the puncta vary in size. Ripley's L statistic was then calculated for each cluster. Significance testing between WT and Crsh were calculated using the 'studperm.test' function of the 'spatstat' package

using the 'Lest' for Ripley's L as the summary function, 'best' for the correction, 0–200 nm for the r interval, three as the minimum number of points, and Tbar for the significance test. Note that one limitation of this analysis is the 'spatstat' package utilizes the binarized images and does not allow for duplicate points, or multiple molecules per pixel. Thus, the number of molecules is not taken into account for these measurements. However, we expect that differences in the Ripley's L statistic are being under-estimated given that WT puncta consist of more molecules per puncta compared to Celsr1$^{Crsh}$.

## Quantification of Frizzled6 and Vangl2 colocalization

Reconstructed SIM images were analyzed for Pearson's R value using 'Coloc 2' in ImageJ Fiji. Entire cell–cell borders between neighboring cells were measured using a segmented ROI.

### Statistical analysis

Statistical analysis was performed in Microsoft Excel or Prism (GraphPad Software, San Diego CA). See figure legends for additional experimental and statistical details. *t*-tests between WT and *Crsh* groups, or as indicated, were performed. p-value significance was defined as follows: *p<0.05, **p<0.01, ***p<0.001, ****p<0.0001.

## Acknowledgements

The authors are grateful to all those who provided resources, technical support, and helpful suggestions that influenced this work. We thank members of the Devenport laboratory for their advice and insightful discussions, especially Katherine Little for reagent generation and manuscript editing, and Dr. Maureen Cetera for critical feedback during manuscript preparation. We thank Gary Laevsky in the Princeton University Confocal Imaging Facility, a Nikon Center of Excellence, for imaging assistance and expertise. We also thank Christina DeCoste and Katherine Rittenbach in the Princeton University Flow Cytometry Resource Facility for assistance with cell sorting. We appreciate those in Princeton University's Laboratory Animal Resources. We kindly thank Dr Elaine Fuchs for the generous donation of the *Celsr1$^{Crsh/+}$* mouse line and the Rutgers Cancer Institute of New Jersey Genome Editing Shared Resource for rederivation services with funding, in part, from the National Cancer Institute of the National Institutes of Health. This work was supported by the National Institute of Arthritis and Musculoskeletal and Skin Diseases, and the Vallee Foundation.

## Additional information

### Competing interests

Danelle Devenport: Reviewing editor, *eLife*. The other authors declare that no competing interests exist.

### Funding

| Funder | Grant reference number | Author |
| --- | --- | --- |
| National Institutes of Health | R01AR068320 | Danelle Devenport |
| National Institutes of Health | R01AR066070 | Danelle Devenport |
| National Institutes of Health | F32AR071764 | Sara N Stahley |
| Vallee Foundation | Scholar Award | Danelle Devenport |

The funders had no role in study design, data collection and interpretation, or the decision to submit the work for publication.

### Author contributions

Sara N Stahley, Conceptualization, Data curation, Formal analysis, Investigation, Visualization, Methodology, Writing - original draft, Writing - review and editing; Lena P Basta, Investigation, Methodology, Writing - review and editing, Performed co-immunoprecipitation experiments and western blot

analysis; Rishabh Sharan, Software, Formal analysis, Methodology, Writing - review and editing, quantitative analyses of dSTORM data; Danelle Devenport, Conceptualization, Supervision, Funding acquisition, Writing - original draft, Writing - review and editing

### Author ORCIDs
Sara N Stahley ⓘ https://orcid.org/0000-0002-2316-8354
Danelle Devenport ⓘ https://orcid.org/0000-0002-5464-259X

### Ethics
Animal experimentation: All procedures involving animals were approved by Princeton University's Institutional Animal Care and Use Committee (IACUC) under protocol #1867. Mice were housed in an AALAC-accredited facility in accordance with the Guide for the Care and Use of Laboratory Animals. This study was compliant with all relevant ethical regulations regarding animal research.

### Decision letter and Author response
Decision letter https://doi.org/10.7554/eLife.62097.sa1
Author response https://doi.org/10.7554/eLife.62097.sa2

## Additional files
### Supplementary files
- Source code 1. MATLAB script for dSTORM puncta analysis.
- Source code 2. MATLAB script for batch processing of dSTORM puncta analysis.
- Source code 3. R script for Ripley's L and K statistics.
- Transparent reporting form

### Data availability
All data generated or analyzed during this study are included in the manuscript and supporting files. Source data files have been provided for Figures 1–8.

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
