## [Decision Letter]

**Acceptance summary:**

This paper examines the role of the transmembrane adhesion protein CELSR-1 in establishing the asymmetric cell-cell junctions in mammalian planar polarized epithelia. CELSR-1 can mediate adhesion between cells, and it also organizes the proteins Fzd6 and Vangl, which are found on opposing membranes in these junctions, into stable clusters, which provides insights into how planar cell polarity develops and maintained. Based on a point mutant in the CELSR-1 extracellular domain, the authors propose that its ability to cluster other PCP proteins is mediated by lateral (cis) interactions of CELSR-1 on each cell surface.

**Decision letter after peer review:**

Thank you for submitting your article "Celsr1 adhesive interactions mediate the asymmetric organization of planar polarity complexes" for consideration by *eLife*. Your article has been reviewed by three peer reviewers, and the evaluation has been overseen by a Reviewing Editor and Anna Akhmanova as the Senior Editor. The reviewers have opted to remain anonymous.

The reviewers have discussed the reviews with one another and the Reviewing Editor has drafted this decision to help you prepare a revised submission.

Summary:

This paper examines the role of CELSR in establishing asymmetric cell-cell junctions in keratinocyte planar cell polarity (PCP). Super-resolution imaging shows that CELSR-1 can organize Fzd6 and Vangl into stable clusters. The clustering is disrupted in the Crsh mutant, which causes defects in PCP and which harbors a single amino acid substitution in the extracellular cadherin repeats of CELSR-1, which the authors interpret as an effect on cis-interactions of CELSR on the cell surface. The segregation of non-junctional Fzd6 and Vangl2 is another potentially important finding for understanding how PCP develops or is maintained. Although clustering has been documented in *Drosophila*, these findings add important new data for the mammalian system.

Essential revisions:

The reviewers find that your data do not establish that the Crsh mutant is specifically defective in cis- vs. trans adhesion, which is a major mechanistic conclusion. The mechanism of CELSR cadherin interactions is not known, and it is possible that the mutant has weakened trans adhesion; unfortunately there do not appear to be mutations equivalent to W2A of classical cadherins that selectively disrupts trans interactions. The failure to see Crsh at borders between keratinocytes or its weak accumulation there in K-562 cells does not distinguish between cis and trans defects, and either kind of defect could explain the FRAP data. The sorting observed in the K-562 adhesion experiments could be due to weakened trans interactions, and/or different levels of surface CELSR molecules. The rescue by induced dimerization likewise could be explained by enhanced avidity of trans interactions. Thus, your results need to be discussed in a balanced manner in which models both of defects in trans as well as cis interactions are considered.

1) Cell surface levels of WT and Crsh CELSR need to be compared (perhaps by cell surface biotinylation) to rule out differences as a possible contribution to the adhesive differences, particularly given the apparently larger amount of intracellular Crsh compared to WT. It may be also be worthwhile to test the artificially dimerized molecules in the K-562 adhesion assay, again, checking cell surface levels.

2) In the text, the authors say that the Crsh mutation does not impair Celsr1 interactions with Vangl2. However, in Figure 8, it appears that the Co-IP of Crsh with Vangl2 is reduced. This should be quantified and commented on in the text. In an IP from FKBP treated Crsh- does this increase Vangl2 co-IP? Comparing colocalization of WT and Crsh mutants with anti-Celsr1 and Vangl2 and Fzd6 antibodies should be done to assess whether the mutant only associates with Fzd6 or at least localizes more with Fzd6 than Vangl2. A related issue occurs when it is stated that "when co-expressed with Fz6-tdTomato or mCherry-Vangl2…in cultured keratinocytes, Celsr1^Crsh^-GFP was still able to drive the redistribution of Vangl2 and Fz6 from the cytoplasm to cell contacts (Figure 8—figure supplement 1)." Only Fz is shown.

[Editors' note: further revisions were suggested prior to acceptance, as described below.]

Thank you for submitting your article "Celsr1 adhesive interactions mediate the asymmetric organization of planar polarity complexes" for consideration by *eLife*. The evaluation of your article has been overseen by a Reviewing Editor and Anna Akhmanova as the Senior Editor.

Summary:

This paper examines the role of CELSR in establishing asymmetric cell-cell junctions in keratinocyte planar cell polarity (PCP). Super-resolution imaging shows that CELSR-1 can organize Fzd6 and Vangl into stable clusters. The clustering is disrupted in the Crsh mutant, which causes defects in PCP and which harbors a single amino acid substitution in the extracellular cadherin repeats of CELSR-1, which the authors interpret as an effect on cis-interactions of CELSR on the cell surface. The segregation of non-junctional Fzd6 and Vangl2 is another potentially important finding for understanding how PCP develops or is maintained. Although clustering has been documented in *Drosophila*, these findings add important new data for the mammalian system.

The authors have carefully addressed the concerns raised in the original reviews, including new data on cell surface expression of Crsh vs. WT and an attempt to quantify co-IP with Fz6.

Essential revisions:

1) In the rebuttal, the authors acknowledge that their data do not decisively establish that the Crsh mutation specifically affects cis interactions, and they note that they want to present this as a working model. While this is true of the Discussion, a minor modification in the Abstract would make clear that the data are not definitive. Specifically, adding "suggesting" to the statement as in "we provide evidence suggesting that a PCP-mutant variant of Celsr1…selectively impairs lateral cis-interactions" (or something along these lines) would help.

2) It would be useful to add the responses regarding non-junctional punctae and Crsh mobility to the text; each of these points can be summarized succinctly and the data shown as supplemental figures.

---

## [Author Response]

Essential revisions:The reviewers find that your data do not establish that the Crsh mutant is specifically defective in cis- vs. trans adhesion, which is a major mechanistic conclusion. The mechanism of CELSR cadherin interactions is not known, and it is possible that the mutant has weakened trans adhesion; unfortunately there do not appear to be mutations equivalent to W2A of classical cadherins that selectively disrupts trans interactions. The failure to see Crsh at borders between keratinocytes or its weak accumulation there in K-562 cells does not distinguish between cis and trans defects, and either kind of defect could explain the FRAP data. The sorting observed in the K-562 adhesion experiments could be due to weakened trans interactions, and/or different levels of surface CELSR molecules. The rescue by induced dimerization likewise could be explained by enhanced avidity of trans interactions. Thus, your results need to be discussed in a balanced manner in which models both of defects in trans as well as cis interactions are considered.

We appreciate the reviewers concerns regarding our conclusions for cis- vs. trans-interactions. We completely agree that without experiments aimed at measuring the strength and/or duration (such as single-molecule biophysics) of trans or cis interactions specifically with respect to Wt and Crsh, we cannot definitely conclude that Crsh is a cis-specific mutant. We are pursuing these experiments as part of a separate study. Within the scope of this current manuscript, we simply wished to present our working model and hypothesis that Crsh is lies within a cis-interacting region of Celsr1. We are open to other possibilities and look forward to fully uncovering this aspect of Celsr biology. To address the reviewers’ concerns, we have made an effort to discuss our results in a more balanced manner, considering models that account for both defects in trans- and cis-interactions. This is reflected in edits to the Abstract, Results and Discussion sections. We have also modified the phrasing in the Discussion section to “propose” a working model rather than stating our findings “support” a model with Crsh defective in cis-interactions.

1) Cell surface levels of WT and Crsh CELSR need to be compared (perhaps by cell surface biotinylation) to rule out differences as a possible contribution to the adhesive differences, particularly given the apparently larger amount of intracellular Crsh compared to WT. It may be also be worthwhile to test the artificially dimerized molecules in the K-562 adhesion assay, again, checking cell surface levels.

We thank the reviewer for this important suggestion. Surface biotinylation was used to measure the cell surface level of Celsr1^WT^ and Celsr1^Crsh^ in the K562 cells used for mixing adhesion assays. These results are now presented in current Figure 2—figure supplement 1, panels C-D. We do find that both Celsr1 forms are expressed at the cell surface at similar levels. Thus, adhesive differences observed with our mixing assay are likely not due to difference in Celsr1 levels at the cell surface. Further, if differences in surface levels were present, we might expect that Wt and Crsh cells would still aggregate but that the lower expressing cells would surround the higher expressing cells, as has been observed for classical cadherins. However, that is not what we see, as the Wt and Crsh really sort into distinct cell aggregates. Even at initial, early time points of mixing, we do not see evidence of the “inner core” / “outer shell” phenotype. This result is now addressed in more detail in the Results section.

2) In the text, the authors say that the Crsh mutation does not impair Celsr1 interactions with Vangl2. However, in Figure 8, it appears that the Co-IP of Crsh with Vangl2 is reduced. This should be quantified and commented on in the text. In an IP from FKBP treated Crsh- does this increase Vangl2 co-IP?

Quantification of the Co-IP experiment is now displayed in current Figure 7, panel C. While we see a slight decrease in Vangl2 interactions compared to Fz6 (both normalized to Celsr1), the results were variable across the three experimental replicates and were not ultimately statistically significant. The text has been modified to account for the quantification addition and we have edited the Results section heading “Crsh does not impair Celsr1 interactions with PCP proteins Frizzled6 and Vangl2” to read “Crsh physically interacts with PCP proteins Frizzled6 and Vangl2”. Our main conclusion from these results are that Crsh is still physically and biochemically able to interact with both Fz6 and Vangl2.

Comparing colocalization of WT and Crsh mutants with anti-Celsr1 and Vangl2 and Fzd6 antibodies should be done to assess whether the mutant only associates with Fzd6 or at least localizes more with Fzd6 than Vangl2.

Colocalization of Celsr1 with either Fz6 and Vangl2 in WT and Crsh mutants is now presented in current Figure 8—figure supplement 1 and 2. We find that Fz6 and Vangl2 colocalization with Celsr1 is similarly reduced in Crsh mutants at vertical borders. Celsr1 and Fz6 colocalization at horizontal borders also resembles that of Crsh at all borders, similar to the findings for Fz6 and Vangl2 colocalization. Interestingly, we do find a statistically significant reduction of Celsr1 colocalization with Vangl2 at horizontal cell borders in Crsh mutants relative to WT. This result, along with the slight reduction (though not statistically significant) in Crsh-Vangl2 interactions by co-IP, does suggest that the Crsh mutation may have a greater effect on Celsr1-Vangl2 interactions.

A related issue occurs when it is stated that "when co-expressed with Fz6-tdTomato or mCherry-Vangl2…in cultured keratinocytes, Celsr1^Crsh^-GFP was still able to drive the redistribution of Vangl2 and Fz6 from the cytoplasm to cell contacts (Figure 8—figure supplement 1)." Only Fz is shown.

For Figure 8—figure supplement 1 (now Figure 7—figure supplement 1), only Fz6 is shown because this experiment with Vangl2 was previously published (Devenport and Fuchs, 2008). The text has been modified to make this clearer to the reader.